# FROM BULK TO BUDGET: BEST PRACTICES TO COMPRESS MULTIMODAL LARGE LANGUAGE MODELS

## ABSTRACT

Multimodal large language models (MLLMs) are increasingly developed to meet diverse deployment needs, varying in scale and computational demand. While recent research has focused on building MLLMs from Small Language Models (SLMs), these efforts remain limited in flexibility and are still data- and compute-intensive. In this paper, we present the first comprehensive study on flexibly compressing and recovering existing MLLMs in a data-efficient manner. Hence, we address a critical gap in the literature by empirically analyzing best practices for adapting to specific hardware or resource limitations. Our study investigates pruning and knowledge distillation techniques, examining their impact on downstream performance across various model compression strategies, including pruning paradigms, recovery training schemes, and data requirements. Key findings reveal that widthwise pruning is particularly effective in resource-constrained scenarios. For smaller compression ratios, finetuning the multimodal projector alone can restore most performance, while combining finetuning with hidden state knowledge distillation proves most effective across all compression levels. Notably, we demonstrate efficient model downsizing using as little as 5% of the original dataset for moderate compression. Our analysis suggests best practices for compressing MLLMs for resource-efficient deployment. With our best practices, Bunny-v1.0-3B retains over 95% of its original performance, while LLaVA-v1.5-7B maintains more than 97%, with compression ratios below 30%.

## 1 INTRODUCTION

State-of-the-art multimodal large language models (MLLMs) (Liu et al., 2023; Chu et al., 2023; Chen et al., 2024b) based on Large Language Models (LLMs) require substantial resources. For instance, the LLaVA family (Liu et al., 2023) includes models with parameter counts ranging from 7 to 34 billion. Even those designed to be more memory-efficient, such as Bunny-v1.0-3B (He et al., 2024), still require significant storage, with 3 billion parameters. Reducing the size of these models without compromising performance is crucial for adapting them to diverse deployment scenarios with varying resource constraints.

Despite the growing need for efficient MLLMs, most existing research has focused on building MLLMs on SLMs (Zhu et al., 2024a; He et al., 2024; Chu et al., 2023). While these approaches successfully reduce the overall model size, their flexibility is constrained by the fixed size of the underlying SLM. Furthermore, training an SLM from scratch to meet desired specifications is computationally expensive (Chu et al., 2023). Meanwhile, efforts to compress multimodal models have largely focused on task-specific tuning (Wang et al., 2023; Shi et al., 2023). To the best of our knowledge, no previous work has investigated general-purpose model compression for MLLMs.

In this work, we aim to uncover the key practices for obtaining effective and compressed MLLMs. We evaluate several techniques designed for compressing LLMs on MLLMs and investigate how different design choices for performance recovery affect the downstream performance of the compressed MLLMs. Our comprehensive empirical study explores several key dimensions: the pruning paradigms applied, the objectives used to restore initial performance, and the amount of data required for effective recovery. Specifically, we examine two MLLMs: a large-scale model (LLaVA-v1.5-7B (Liu et al., 2024)) and a model already optimized for efficiency (Bunny-v1.0-3B (He et al., 2024)).

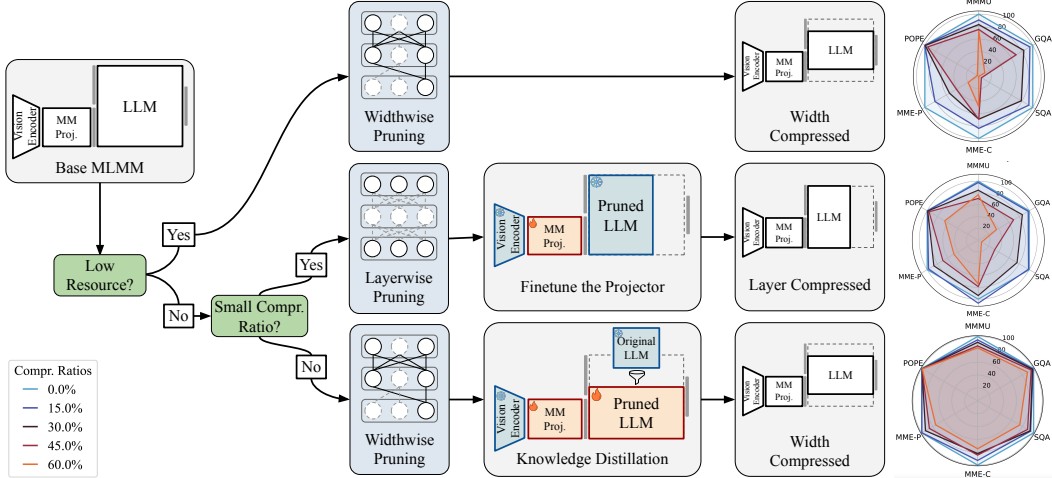

Figure 1: Overview of our best practices for MLLM compression. After evaluating two pruning strategies—widthwise and layerwise pruning—and multiple recovery strategies, we propose distinct compression approaches for MLLMs. The decision flow highlights the scenarios for applying each method, depending on resource availability and compression ratio requirements. For each approach, we display the resulting compressed model performance across a range of compression ratios (0-60%) on a set of multimodal benchmarks. The spider plots illustrate the retained performance across different tasks, demonstrating each strategy's effectiveness at various compression levels.

We assess their performance on visual question answering (Hudson & Manning, 2019; Lu et al., 2022) and instruction-following tasks (Li et al., 2023; Yin et al., 2023).

Given that the LLM contains the majority of the parameters (95% for LLaVA-v1.5-7B, 86% for Bunny-v1.0-3B), we compress it by applying two distinct pruning strategies: layerwise pruning, which removes entire transformer layers, and widthwise pruning, which reduces the number of attention heads and MLP hidden dimensions. We then evaluate different strategies to recover the potential performance loss: supervised finetuning, knowledge distillation from the original model using Kullback-Leibler divergence or reversed Kullback-Leibler divergence on the logits distribution or L2 loss on the intermediate features. Finally, we investigate how to combine these losses and their effectiveness w.r.t. the amount of available training data.

Our systematic evaluation across different compression ratios led to different key findings:

- **Widthwise pruning is more effective in low-resource scenarios** as it produces an efficient model even without recovery training.

- **With recovery training, layerwise pruning is better for small ratios**, while widthwise pruning usually outperforms it at larger ones (greater than 40%).

- **Finetuning only the multimodal projector is sufficient at small compression ratios**, where pruning has a minimal impact on the language model itself but destroys the multimodal alignment.

- **The best recovery strategy is supervised finetuning coupled with intermediate representation distillation**, consistently achieving the the highest performance across all compression ratios.

- **The higher the pruning ratio, the higher the amount of data needed for recovering the performance**. While with small ratios (less than 50%) even 5% of the data might suffice, this quantity increases for larger ones.

We highlight our key findings in Figure 1. These findings consitute a set of best practices that practitioners can follow when compressing MLLMs and researchers can consider for developing effective pruning strategies. To ease their exploitation and future studies, we will release our codebase, benchmark as well as the compressed model checkpoints upon acceptance.

## 2 RELATED WORK

**Pruning.** Unstructured pruning (Dong et al., 2017; Frankle & Carbin, 2019; Lee et al., 2020; Park et al., 2020; Sanh et al., 2020b; Farina et al., 2024) removes individual weights or neurons. While such approaches can achieve strong compression rates with minimal accuracy trade-offs, they usually require specialized hardware or software for effective acceleration. Structured pruning (Ding et al., 2019; Li et al., 2017; Liu et al., 2021; You et al., 2019) eliminates entire groups of parameters to reduce both the model's size and its computational overhead. Semi-structured pruning offers a middle ground between structured and unstructured methods by selectively pruning certain model parts. In the context of LLMs, Fang et al. (2023) and Ma et al. (2023) have successfully applied structured pruning, achieving significant sparsity with minimal performance degradation. Xia et al. (2024) targets transformer layers and demonstrates that some layers can be pruned more aggressively without compromising accuracy. Meanwhile, Dery et al. (2024) propose a dynamic pruning strategy that adjusts pruning throughout the training process. In this work, we focus on structure pruning as well as the recovery strategies for MLLMs.

**Further Compression Methods.** In addition to pruning, techniques like quantization and low-rank factorization are also widely used for model compression. Quantization (Bai et al., 2021; Yao et al., 2022; Zafrir et al., 2019) reduces model size and computational cost by lowering the precision of model parameters, enabling efficient inference with minimal performance loss. Low-rank factorization (Hsu et al., 2022; Hu et al., 2021b; Lan et al., 2020; Ashkboos et al., 2024) compresses models by approximating large weight matrices through the product of smaller matrices, effectively reducing the number of parameters while maintaining most of the model's capacity. While these methods can offer significant compression, we focus on pruning techniques, which allow for more granular control over the architecture by directly targeting and removing redundant components.

**Knowledge distillation (KD)** (Hinton et al., 2015) is a standard method for compressing LLMs by transferring knowledge from a large teacher model to a smaller student model (Gou et al., 2021; Sanh et al., 2020a). In NLP classification settings, KD is often applied by having the student model replicate the teacher's output distribution (Liang et al., 2021; Song et al., 2020; Zhang et al., 2023), hidden states Jiao et al. (2020); Sun et al. (2019), or attention patterns (Wang et al., 2020; 2021), allowing the student to learn from the teacher's internal representations effectively. For text generation tasks, Xu et al. (2024) provides a comprehensive survey of the role of knowledge distillation in language models. Hsieh et al. (2023) introduce multi-stage distillation, transferring intermediate representations to help the student model capture more detailed features. Gu et al. (2023) propose to replace the forward Kullback-Leibler divergence with a reverse Kullback-Leibler divergence to prevent the student model from overestimating the low-probability regions of the teacher distribution.

**Efficient MLLMs.** Recent studies (Jin et al., 2024; Zhu et al., 2024b; Lin et al., 2024; Wei et al., 2024) have explored Multimodal Small Language Models (MSLMs). Models such as LLaVA-Phi (Zhu et al., 2024b) utilize pretrained small language models to lower computational costs, while MobileVLM (Chu et al., 2023) concentrates on projector designs to enhance MSLM performance. The Bunny model (He et al., 2024) explores the effects of training data size on performace. Although these approaches reduce model size, they are constrained by the fixed dimansions of the base SLM. Our study specifically addresses methods for customizing the size of existing MLLMs through structured pruning and recovery strategies.

While most multimodal structured compression efforts, such as EfficientVLM (Wang et al., 2023) and UPOP (Shi et al., 2023), focus on task-specific tuning for tasks like visual question answering and image captioning, general-purpose model compression for MLLMs remains underexplored. Our work addresses this gap by investigating structured pruning techniques and recovery strategies applicable across a variety of multimodal tasks. Unlike previous approaches optimizing models for specific tasks, our study provides general-purpose compression guidelines for MLLMs.

## 3 METHODOLOGY

This section outlines our approach to compressing MLLMs. We first introduce two pruning strategies: layerwise and widthwise pruning. We then describe methods to recover model performance through supervised finetuning and knowledge distillation.

**Notation.** Given a triplet $\mathbf{X} = \{\mathbf{x}_v, \mathbf{x}_p, \mathbf{x}_r\}$, the objective of an MLLM $m_\theta$, parameterized by $\theta = \{\psi, \phi, \mathbf{W}\}$, is to generate a response $\mathbf{x}_r$ based on an input image $\mathbf{x}_v$ and a text prompt $\mathbf{x}_p$, such that $m_\theta(\mathbf{x}_v, \mathbf{x}_p) = \mathbf{x}_r$. The MLLM typically consists of a vision encoder $g_\psi(\cdot)$, an LLM $f_\phi(\cdot)$, and a multimodal projector $\mathbf{W}$ aligning the two modalities. The prompt $\mathbf{x}_p$ is tokenized into $\mathbf{T}_p$, while the vision encoder processes the image $\mathbf{x}_v$ to extract visual features, which are then converted into language embedding tokens $\mathbf{T}_v$ via the multimodal projector:

$$\mathbf{T}_v = \mathbf{W} \cdot g_\psi(\mathbf{x}_v) \ \text{ and } \ f_\phi(\mathbf{T}_v \odot \mathbf{T}_p) = \mathbf{x}_r. \tag{1}$$

The concatenated visual tokens $\mathbf{T}_v$ and prompt tokens $\mathbf{T}_p$ are fed into the LLM's $M$ layers, producing hidden states $\{\mathbf{H}_i \in \mathbb{R}^{T \times d}\}_{i=1}^M$, where $T$ is the number of tokens and $d$ is the hidden dimension. Finally, the probabilities $p_{m_\theta}(\mathbf{x}_r | \mathbf{x}_v, \mathbf{x}_p, \tau)$ are computed by passing the final hidden state through the classification head with softmax temperature $\tau$.

### 3.1 PRUNING

Pruning seeks to reduce the number of parameters in a model, thus decreasing its computational cost. In MLLMs, the majority of parameters $\theta$ are concentrated in the LLM $f_\phi$, so downsizing it can significantly reduce the overall computational burden. The LLM is typically structured along two main dimensions: depth and width. Depth refers to the number of stacked transformer layers, while width pertains to the internal structure of each layer, including the multi-head attention mechanism and the multi-layer perceptron (MLP). In this paper, we explore two pruning strategies to reduce the parameter count in LLMs: layerwise pruning, which removes entire transformer layers, and widthwise pruning, which eliminates the least important components within each layer.

To determine which layers or components to prune, we draw a small subset of $n$ samples from the the original visual instruct-tuning dataset as the calibration dataset $\mathcal{D} = \{\mathbf{x}_v^j, \mathbf{x}_p^j, \mathbf{x}_r^j\}_{j=1}^n$. The importance of each layer or component is assessed, and those with the lowest importance are pruned.

#### 3.1.1 LAYERWISE PRUNING

Empirical research (Fan et al., 2019; Sajjad et al., 2023) has shown that large transformer models often contain redundant layers, allowing several to be removed with minimal impact on accuracy. To identify and remove these redundant layers, we use the Block Influence (BI) score (Men et al., 2024), which quantifies the importance of layer $i$ through the cosine distance between input $\mathbf{H}_i$ and output hidden states $\mathbf{H}_{i+1}$. The key assumption is that layers that cause larger changes in hidden states have a greater influence on model performance. The BI score of layer $i$ is then calculated by

$$\text{BI}_i(\mathcal{D}) = 1 - \mathbb{E}_{\mathbf{X} \sim \mathcal{D}, t} \left[ \frac{\mathbf{H}_{i,t}^\mathsf{T} \mathbf{H}_{i+1,t}}{\|\mathbf{H}_{i,t}\|_2 \|\mathbf{H}_{i+1,t}\|_2} \right], \tag{2}$$

where $\mathbf{H}_{i,t}$ represents the $t^{th}$ row of $\mathbf{H}_i$. After calculating the BI scores, the layers are ranked by importance, and those with the lowest scores are pruned.

#### 3.1.2 WIDTHWISE PRUNING

Previous research has shown that transformer layers also exhibit width redundancy, meaning that only a subset of attention heads (Voita et al., 2019; Michel et al., 2019) or MLP dimensions (Mc-Carley et al., 2019; Hudson & Manning, 2019) are critical for model performance. To address this, we apply dependency-based structural pruning, which removes redundant widthwise components while minimizing the impact on the model's performance. Specifically, we first identify groups of interdependent structures and then prune entire groups based on their collective importance.

Following the methods of Fang et al. (2023) and Ma et al. (2023), we begin by constructing dependency relationships within each LLM layer. Let $N_i$ and $N_j$ represent two neurons in the layer, where $\text{In}(N_i)$ and $\text{Out}(N_i)$ represent the neurons connected to $N_i$ as inputs and outputs, respectively. The dependency of neuron $N_j$ on $N_i$ is defined as:

$$N_j \in \text{Out}(N_i) \cap \text{Num}_{\text{In}(N_j)} = 1, \text{ or } N_j \in \text{In}(N_i) \cap \text{Num}_{\text{Out}(N_j)} = 1, \tag{3}$$

where $\text{Num}_{\text{In}(N_i)}$ refers to the number of nodes connected to $N_i$ as inputs and $\text{Num}_{\text{Out}(N_j)}$ is the number of nodes connected to $N_j$ as outputs. If neuron $N_i$ is pruned, all its dependent neurons $N_j$

must also be pruned. This process results in a set of dependency graphs $\mathcal{G} = \{w_i^k\}_{i=1}^M$, where $M$ is the number of structures in the graph and $w_i^k$ represents the $k^{th}$ weight parameter within a structure.

Once the dependency graphs are constructed, we assess their importance at the group level, since all weights within a graph must be pruned together. Group importance is evaluated by comparing the vision-language modeling loss $\mathcal{L}_{CE}(m_\theta(\mathbf{x}_v, \mathbf{x}_q), \mathbf{x}_r)$ on the calibration dataset, both with and without the weight. To efficiently approximate the importance, we apply a Taylor expansion using gradient information. The importance function is given by:

$$I_{w_i^k}(\mathbf{X}) = |\mathcal{L}_{CE}(\mathbf{X}, m_\theta) - \mathcal{L}_{CE}(\mathbf{X}, m_\theta^{w_i^k=0})| \approx \left| \frac{\partial \mathcal{L}_{CE}(\mathbf{X}, m_\theta)}{\partial w_i^k} w_i^k \right|. \quad (4)$$

We then prune the graphs with the lowest group importance $I_\mathcal{G}$:

$$I_\mathcal{G}(\mathcal{D}) = \mathbb{E}_{\mathbf{X} \sim \mathcal{D}} \left[ \sum_i^M \sum_k I_{w_i^k}(\mathbf{X}) \right]. \quad (5)$$

### 3.2 Recovery Training

Pruning a large multimodal language model results in performance degradation, affecting both language modeling and cross-modality alignment. To mitigate this, we investigate two recovery training methods: supervised finetuning (Sec. 3.2.1) and knowledge distillation (Sec. 3.2.2). We consider the orginal teacher model $m_\theta^\mathrm{T}$, the pruned student model $m_{\theta'}^\mathrm{S}$, and a recovery dataset $\mathcal{D}$.

#### 3.2.1 Recovery Training with supervised finetuning

A simple yet effective approach to recovery training is supervised finetuning on the original dataset. This method helps counteract performance degradation by allowing the model to adapt its parameters to the modified architecture while taking advantage of the detailed annotations in the original dataset. Here, we first focus on training only the multimodal projector to realign the vision and language spaces. Second, we jointly finetune both the projector and the language model while keeping the vision encoder fixed, as finetuning the vision encoder does not improve performance (Karamcheti et al., 2024). We use the cross-entropy loss for supervised finetuning, denoted as

$$\mathcal{L}_{sft}(m_{\theta'}^\mathrm{S}, \mathcal{D}) = \mathbb{E}_{\mathbf{X} \sim \mathcal{D}}[\mathcal{L}_{CE}(m_{\theta'}^\mathrm{S}(\mathbf{x}_v, \mathbf{x}_p), \mathbf{x}_r)]. \quad (6)$$

#### 3.2.2 Recovery Training with knowledge distillation

Knowledge distillation (KD) is a method used to transfer knowledge from a large, well-trained model (the teacher) to a smaller or pruned model (the student) (Hinton et al., 2015). This approach allows the pruned model to regain lost performance by mimicking the decision-making process of the more capable teacher. In our setup, the uncompressed model acts as the teacher, while the pruned model serves as the student. We explore two main strategies, logits-based KD and hidden state based KD, evaluating different loss functions and their trade-offs.

**Logits-based KD** focuses on aligning the output probability distributions of the pruned model with those of the teacher model. The logits-based KD loss is defined as

$$\mathcal{L}_{logits}(m_{\theta'}^\mathrm{S}, m_\theta^\mathrm{T}, \mathcal{D}) = \mathbb{E}_{\mathbf{X} \sim \mathcal{D}} \left[ \mathcal{L}_{KD}(p_{m_\theta^\mathrm{T}}(\mathbf{x}_r|\mathbf{x}_v, \mathbf{x}_p, \tau), p_{m_{\theta'}^\mathrm{S}}(\mathbf{x}_r|\mathbf{x}_v, \mathbf{x}_p, \tau)) \right]. \quad (7)$$

We leverage two distinct KD losses to evaluate the differences between these logit distributions. Given the teacher distribution $p_\theta$ and the student distribution $q_{\theta'}$, the standard KD objective minimizes the approximated forward Kullback–Leibler (KL) divergence between these two distributions, denoted as $\mathcal{L}_{KD}(p_\theta, q_{\theta'}) = D_{\mathrm{KL}}(p_\theta \| q_{\theta'})$. This approach encourages the student distribution to match all the modes of the teacher distribution.

However, minimizing forward KL can lead $q_\theta$ to assign excessively high probabilities to areas where $p$ has little or no probability mass (Malinin & Gales, 2019). In contrast, Reversed Kullback–Leibler divergence (RKL) minimizes $\mathcal{L}_{KD}(p_\theta, q_{\theta'}) = D_{\mathrm{KL}}(q_{\theta'} \| p_\theta)$, encouraging $q_{\theta'}$ to focus on the major modes of $p_\theta$ while assigning low probabilities to its less significant regions. This helps the student model avoid learning unnecessary long-tail variations of the teacher distribution and instead focus on generating more accurate responses (Gu et al., 2023; Holtzman et al., 2019).

**Hidden State Matching** involves aligning the pruned model's intermediate representations (hidden states) $\mathbf{H}_i^{m_{\theta'}^{\mathrm{S}}}$ with the teacher model's $\mathbf{H}_i^{m_\theta^{\mathrm{T}}}$. The corresponding loss for a layer $i$ can be defined as

$$\mathcal{L}_{match}(m_{\theta'}^{\mathrm{S}}, m_\theta^{\mathrm{T}}, \mathcal{D}) = \mathbb{E}_{\mathbf{X} \sim \mathcal{D}} \left[ \mathcal{L}_{feat}(\mathbf{H}_i^{m_{\theta'}^{\mathrm{S}}}, \mathbf{H}_i^{m_\theta^{\mathrm{T}}}) \right], \tag{8}$$

where $\mathcal{L}_{feat}$ refers to a feature matching loss. Both Yang et al. (2024) and Popp et al. (2024) suggest that applying a feature-based L2 distillation loss improves the student model's performance, particularly for pre-trained vision-language models. Consequently, we employ L2 loss as the feature matching loss $\mathcal{L}_{feat} = \| \cdot - \cdot \|_2^2$.

The total loss for recovery training is computed as:

$$\mathcal{L}(m_\theta^{\mathrm{S}}, m_\theta^{\mathrm{T}}, \mathcal{D}) = \alpha \mathcal{L}_{sft}(m_{\theta'}^{\mathrm{S}}, \mathcal{D}) + \beta \mathcal{L}_{logits}(m_{\theta'}^{\mathrm{S}}, m_\theta^{\mathrm{T}}, \mathcal{D}) + \gamma \mathcal{L}_{match}(m_{\theta'}^{\mathrm{S}}, m_\theta^{\mathrm{T}}, \mathcal{D}) \tag{9}$$

where $\alpha$, $\beta$, and $\gamma$ are the coefficients that balance the contributions of the different loss components.

## 4 EXPERIMENTS

In this section, we first introduce our experimental setup and then demonstrate the main findings on model pruning (Sec. 4.1) and performance recovery (Sec. 4.2, Sec. 4.3). Finally, we highlight the findings on recovery training using only a small fraction of data (Sec. 4.4).

**Experimental setup.** We evaluate pruning and knowledge distillation strategies on both a large-scale MLLM model (LLaVA-v1.5-7B (LLaVA) (Liu et al., 2024)) and a smaller-scale MLLM model (Bunny-v1.0-3B (Bunny) (He et al., 2024)). LLaVA is built upon Vicuna-v1.5 (Chiang et al., 2023) with 6.7 billion parameters, and Bunny is based upon Phi-2 (Javaheripi et al., 2023) with 2.8 billion parameters. We provide a detailed overview of the model architectures in Appendix A. For both models, we exclusively use their visual instruction tuning datasets: LLaVA-v1-5-mix665k (Liu et al., 2024) for LLaVA and Bunny-695K (He et al., 2024) for Bunny. During pruning, we randomly select 10 samples as the calibration dataset to compute layer importance. For recovery training, we experiment with various portions of the original dataset (i.e. 5%, 10%, 20%, and 100%) for finetuning and knowledge distillation. We set the distillation temperature to 2.0 for logits-based distillation and use the final layer representation for hidden state matching (see Appendix B).

We evaluate the pruned and recovery-trained models on visual question-answering tasks using GQA (Hudson & Manning, 2019) and SQA-I (Lu et al., 2022), as well as instruction-following tasks with POPE (Li et al., 2023), MME-Cognition, MME-Perception (Yin et al., 2023) and MMMU (Yue et al., 2024). To ensure consistency, we use the lmms-eval suite (Bo et al., 2024) for all evaluations. For clearer comparisons, we calculate the relative performance as a percentage of the original (uncompressed) model's performance on each benchmark.

### 4.1 THE EFFECT OF PRUNING ON THE MODEL PERFORMANCE AND RESOURCES USAGE

In this section, we begin by exploring the techniques to obtain the best pruned model. We then examine the resulting reductions in memory usage and computational requirements.

**Comparison of Pruning Techniques.** As illustrated in Figure 2 (blue lines), for both the Bunny and LLaVA model, widthwise pruning consistently outperforms layerwise pruning in terms of model performance after pruning without recovery training. Specifically, for small compression ratios, such as 15%, the Bunny model retains 95% of its performance, while LLaVA retains 93% (see Appendix C for full results). In resource-constrained scenarios, widthwise pruning without recovery training offers an efficient strategy when a small compression ratio is required. However, as the compression ratio increases, both widthwise and layerwise pruned models show significant performance degradation. Overall, widthwise pruning better preserves the model's structure and information flow, allowing it to keep performance with minimal adjustments, especially at lower compression ratios.

The impact of the pruning method on model performance after finetuning both the projector and the LLM is also illustrated in Figure 2 (green lines). For smaller compression ratios (less than 40%), layerwise pruning offers a slight advantage over widthwise pruning, while widthwise pruning

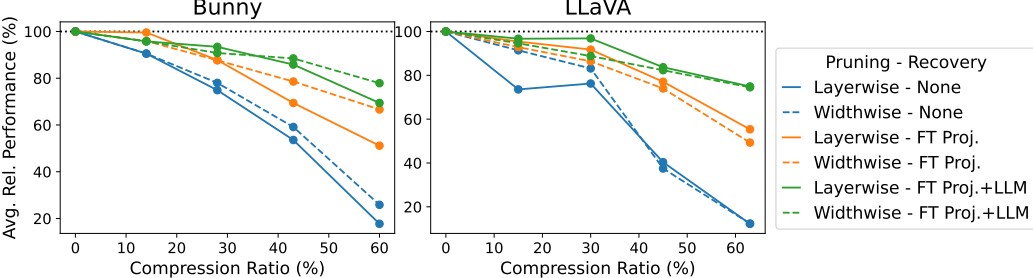

Figure 2: Comparison of pruning and finetuning strategies on two MLLMs, Bunny and LLaVA. The plot shows the average relative performance under three scenarios: pruning only, pruning followed by finetuning the projector, and pruning followed by finetuning both the projector and the LLM. For smaller compression ratios, finetuning only the projector effectively recovers performance. For larger compression ratios, finetuning the projector and the LLM leads to better recovery, indicating the need for broader adjustments as more parameters are pruned.

delivers better overall performance for larger compression ratios (greater than 40%). This suggests that finetuning plays a crucial role in reconstructing inter-layer connections and reoptimizing layer components.

*Best Practice for MMLM Pruning.* Widthwise pruning generally proves more effective than layerwise pruning in obtaining the best pruned model. A widthwise pruned model can often be deployed without recovery training when targeting a small compression ratio (less than 20%). Regarding post-finetuning performance, layerwise pruning shows a slight advantage at compression ratios below 30%, whereas widthwise pruning performs marginally better at higher compression ratios.

**From compression ratio to resource usage.** Table 1 provides an overview of how different compression ratios impact memory usage and FLOPS for both the Bunny and LLaVA models compressed via widthwise pruning. Memory consumption refers to the allocated GPU memory, while FLOPS are measured using the Calflops codebase[1]. The results demonstrate that higher compression ratios consistently lead to both memory and compute reductions. For example, at a 30% compression ratio, we observe a memory reduction of 25% for Bunny and 28% for LLaVA, with a corresponding de-

Table 1: Memory requirements (Mem.) and FLOPS for the Bunny and LLaVA models at various compression ratios. The models are pruned widthwise. The evaluation is performed in inference mode, where each model is provided with an image and a prompt containing 50 tokens.

| Ratio | Bunny | | LLaVA | |
|---|---|---|---|---|
| | Mem. (MiB) | FLOPS (T) | Mem. (MiB) | FLOPS (T) |
| 0% | 6,167 | 4.77 | 13,546 | 9.57 |
| 15% | 5,380 | 4.14 | 11,530 | 8.21 |
| 30% | 4,597 | 3.50 | 9,548 | 6.89 |
| 45% | 3,770 | 2.84 | 7,470 | 5.49 |
| 60% | 2,992 | 2.20 | 5,435 | 4.17 |

crease in FLOPS of 27% for both models. These reductions continue to scale with larger compression ratios; at a 60% compression ratio, memory usage and FLOPS decrease by 50-60%. We observe similar results for layerwise pruning (see Appendix D). This indicates that the achieved compressions directly translate into improvements in memory efficiency and computational cost.

## 4.2 SUPERVISED FINETUNING FOR PERFORMANCE RECOVERY AFTER PRUNING

Compressing the LLM can impair its language modeling capabilities. Additionally, whether pruning the LLM decoder would disrupt the alignment between vision and language remains underexplored. To explore these issues, we experiment with two approaches: finetuning only the multimodal projector and jointly finetuning both the projector and the LLM. Following the previous research (Karamcheti et al., 2024), which shows that training the vision encoder may degrade overall model performance, we keep the vision encoder frozen in both setups. To facilitate fast recovery, we employ the low-rank approximation, LoRA (Hu et al., 2021a), while finetuning the LLM.

**Finetuning the multimodal projector.** As shown in Figure 2 (orange lines), finetuning the multimodal projector significantly restores performance. At lower compression ratios (less than 20%), finetuning only the projector achieves results comparable to jointly finetuning the LLM. For both

---

[1]Caflops codebase: https://github.com/MrYxJ/calculate-flops.pytorch

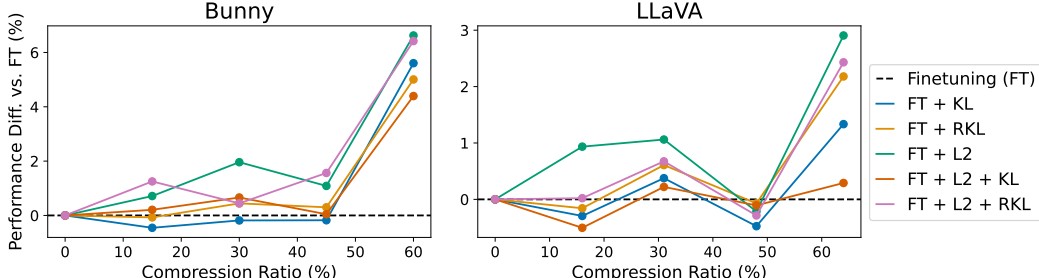

Figure 3: Comparison of different distillation recovery strategies (KL loss, RKL loss, L2 loss, and their combinations) for Bunny and LlaVA models pruned with widthwise pruning. The plot shows the relative performance improvement of each strategy over standard finetuning across various compression ratios. The results demonstrate that distillation helps recover more performance than finetuning alone, with the L2 loss component consistently leading to the largest performance gains.

Bunny and LLaVA, finetuning the projector retains at least 95% of the performance at a compression ratio of 15%. As the compression ratio increases, the loss of language modeling ability becomes more pronounced, making projector-only finetuning insufficient to recover the model's performance fully. Nevertheless, even at a compression ratio of 60%, only finetuning the multimodal projector can still recover 60 to 80% of the performance by realigning the vision and language inputs. This shows that pruning specific LLM structures in the MLLM can both damage the language modeling ability and introduce modality misalignment, making the model incapable of comprehending vision.

**Finetuning both the projector and the LLM.** While a significant portion of the recovered performance is due to realigning the visual and textual inputs, we observe consistent gains from additionally finetuning the pruned LLM (green lines in Figure 2), especially at higher compression ratios (greater than 40%). This indicates that the pruned model not only suffers from modality misalignment but also experiences a decline in its language modeling capabilities. We can partly restore these lost capabilities by finetuning the LLM. At a compression ratio of 40%, finetuning both the projector and the LLM restores more than 80% of the original model's performance. Even at a compression ratio of 60%, finetuning recovers close to 80% of the model's original performance.

*Best Practice for Supervised Finetuning.* When a small compression ratio of around 15% is required, finetuning the multimodal projector alone is typically sufficient to recover most of the model's performance. For higher compression ratios (greater than 40%), incorporating finetuning of the LLM yields additional performance improvements.

## 4.3 KNOWLEDGE DISTILLATION FOR PERFORMANCE RECOVERY AFTER PRUNING

This section investigates the impact of combining knowledge distillation with finetuning. We perform ablation studies by adjusting the weights of each loss component to evaluate their individual contributions (see 3.2.2). Additional ablation results are in the Appendix E.

As shown in Table 2 for the Bunny model compressed with layerwise pruning, we compare a logit-based approach (RKL) and a hidden state matching strategy (L2), both with and without a finetuning loss component. The results demon-

Table 2: Comparison of distillation strategies with and without finetuning for the Bunny model compressed via layerwise pruning. Finetuning helps stabilize performance and prevents *model collapse*, especially at higher compression ratios.

| Ratio | Bunny | | | | |
| | FT | L2 | L2+FT | RKL | RKL+FT |
|---|---|---|---|---|---|
| 15% | 96.30% | 95.51% | **99.59%** | 96.88% | **98.70%** |
| 30% | 94.33% | 88.13% | **95.03%** | 92.21% | **93.81%** |
| 45% | 86.70% | *56.96%* | **90.19%** | 82.57% | **88.50%** |
| 60% | 69.38% | *47.61%* | **72.62%** | *12.61%* | **69.85%** |

strate that incorporating a supervised finetuning loss significantly enhances and stabilizes distillation performance. For example, when applying only the distillation loss, the L2 and RKL methods can recover 85% of the original performance at a compression ratio of 30%. However, for higher compression ratios, adding the finetuning loss becomes critical in preventing model collapse. At a compression ratio of 60%, combining the finetuning loss with distillation dramatically improves performance—L2 distillation increases from 47.61% to 72.62% and RKL distillation from 12.61%

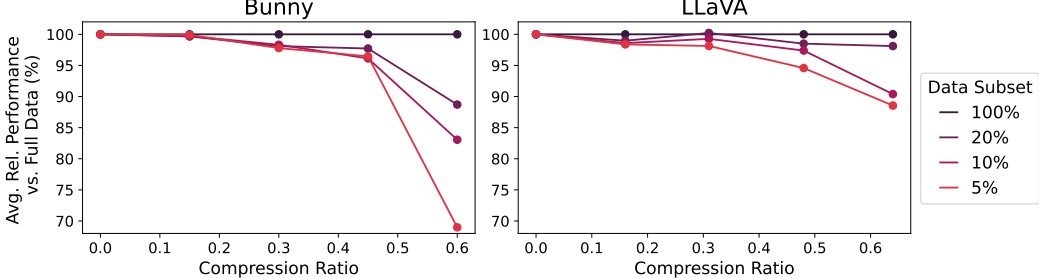

Figure 4: Comparison of recovery performance using different percentages of training data (100%, 20%, 10%, and 5%) for finetuning and distillation after pruning across Bunny and Llava models. For smaller compression ratios, even a small percentage of the training data (as low as 5%) is sufficient to recover most of the original performance. However, as the compression ratio increases, more training data is required to achieve higher recovery performance.

to 69.85%. This pattern is consistently observed across all models and compression techniques evaluated. While knowledge distillation alone can partially recover performance after pruning, its effectiveness is limited without the integration of finetuning.

Figure 3 compares various distillation strategies based on their relative improvement over finetuning alone when widthiwse pruning is applied (see Appendix F for further results on layerwise pruning). Our results indicate that applying the L2 loss to align the hidden states of the student and teacher in the final layer yields the best performance, or at least matches other methods. Unlike logit-based approaches, which require the student to replicate the teacher's output distribution, the L2 loss method enables the student to better capture the teacher's feature representations directly, leading to enhanced performance. Additionally, we observe that RKL generally outperforms KL across most compression ratios, a result consistent with the findings of Gu et al. (2024).

***Best Practice for Knowledge Distillation.*** Knowledge distillation, particularly when combined with finetuning and using L2 loss to map the intermediate states, delivers the most effective performance recovery after pruning across all compression ratios.

## 4.4 DATA EFFICIENT RECOVERY

In this section, we investigate the feasibility of performing recovery training using only a small fraction of the available data. Figure 4 shows the models' performance after recovery training with different portions of the original dataset relative to training with the full 100%. Both models undergo widthwise pruning and recovery training incorporating RKL and L2 loss functions. Remarkably, for

Table 3: Performance of the best compressed models. The size is the number of total parameters of the model, while the ratio, short for compression ratio, indicates the proportion of remaining LLM parameters compared to the pre-pruning state. When the compression ratio (Ratio) is below 40%, we apply depthwise pruning. For ratios above 40%, we use widthwise pruning. During the recovery phase, we employ supervised finetuning combined with L2 loss to match the hidden states. For both Bunny and LLaVA, 95% performance is retained if the compression ratio is smaller than 40%.

| Method | Size | Ratio | MMMU | GQA | SQA | MME-C | MME-P | POPE | AVG | AVG-% |
|---|---|---|---|---|---|---|---|---|---|---|
| **Bunny-v1.0-3B** | | | | | | | | | | |
| | 3.2B | 0% | 34.10 | 54.72 | 70.70 | 289.30 | 1487.71 | 87.82 | 59.65 | 100.00% |
| Depth+FT+L2 | 2.8B | 15% | 33.00 | 54.56 | 70.00 | 304.29 | 1457.06 | 87.97 | 59.40 | 99.59% |
| Depth+FT+L2 | 2.5B | 30% | 32.30 | 53.08 | 68.12 | 252.50 | 1349.91 | 87.53 | 56.68 | 95.03% |
| Width+FT+L2 | 2.0B | 45% | 29.10 | 52.31 | 63.06 | 244.64 | 1281.66 | 87.09 | 54.37 | 91.15% |
| Width+FT+L2 | 1.6B | 60% | 28.10 | 48.72 | 53.20 | 216.07 | 1115.33 | 86.73 | 49.92 | 83.69% |
| **LLaVA-v1.5-7B** | | | | | | | | | | |
| | 7.0B | 0% | 35.10 | 61.98 | 68.67 | 363.21 | 1511.33 | 86.99 | 62.28 | 100.00% |
| Depth+FT+L2 | 6.3B | 15% | 36.40 | 61.20 | 68.42 | 337.86 | 1442.35 | 86.94 | 61.22 | 98.29% |
| Depth+FT+L2 | 5.5B | 30% | 36.00 | 60.34 | 68.82 | 318.57 | 1496.60 | 85.98 | 60.96 | 97.88% |
| Width+FT+L2 | 3.8B | 45% | 30.80 | 57.74 | 52.90 | 215.00 | 1191.17 | 85.74 | 52.27 | 83.92% |
| Width+FT+L2 | 2.8B | 60% | 27.70 | 52.32 | 46.26 | 211.79 | 1085.97 | 84.06 | 48.52 | 77.90% |

compression ratios below 50%, using just 5% of the original data is sufficient to achieve over 95% of the performance compared to using the full dataset. However, as the compression ratio increases, the amount of data required for effective recovery training also grows. For a compression ratio of 60%, the relative performance drops below 90% for LLaVA and diminishes even further to below 70% for Bunny. Nevertheless, using only a small portion of the training data appears to be a valid option, significantly lowering the required time and cost for compressing and finetuning MLLMs.

***Best Practice for Data Efficient Recovery.*** At small to medium compression ratios less than 50%, using just 5% of the dataset is enough to achieve performance comparable to full data training. However, for compression ratios greater than 50%, full data training becomes necessary to recover performance effectively.

### 4.5 MODEL COMPRESSION RESULTS FOLLOWING OUR BEST PRACTICES

In this section, we summarize our key findings as a set of best practices and highlight model performance achieved by following them. Based on the empirical results from the previous section, we outline the following best practices for compressing MLLMs:

> **Best Practices for MLLM Compression and Recovery**
>
> - **Widthwise pruning is more effective in low-resource settings**, yielding an efficient model even without the need for recovery training.
> - **With recovery training**, layerwise pruning excels for smaller compression ratios (below 40%), while widthwise pruning performs better at higher ratios (above 40%).
> - **For small compression ratios**, fine-tuning just the multimodal projector is often sufficient to restore performance, with minimal impact from pruning.
> - **For recovery training**, combining finetuning with knowledge distillation of the intermediate representations using L2 loss consistently achieves the highest performance across all compression ratios.
> - **Data efficiency** can be significantly boosted, requiring only 5% of the original data to match full-data training results, though full datasets are still needed for high compression ratios.

These guidelines provide insights for researchers aiming to develop new techniques for deploying MLLMs, enabling more effective model customization for specific deployment needs.

To illustrate the performance at different compression ratios, Table 3 offers a detailed comparison of results for both Bunny and LLaVA across various multimodal benchmarks. The results show that, with compression ratios below 30%, Bunny retains over 95% of its original performance, while LLaVA maintains more than 97%. Even at higher compression ratios, up to 60%, our best practices preserve an average performance of 83% for Bunny and 78% for LLaVA. These findings underscore the feasibility of compressing MLLMs without incurring significant performance degradation. We provide qualitative results for the compressed models in Appendix G. Additionally, other compression techniques, such as quantization, could be layered on top of the current framework to further reduce inference time and memory usage, see Appendix H. We also include InternVL (Chen et al., 2024a) in Appendix I to show our best practices generalize to other model.

## 5 CONCLUSION

In this work, we investigated efficient compression techniques for MLLMs, focusing on two key pruning strategies: width and layerwise. Our study assessed the impact of these strategies on model performance, both before and after recovery training, across various compression ratios. We further explored recovery methods such as supervised finetuning and knowledge distillation to address performance degradation caused by pruning. We formulate our findings as best practices, which offer practical guidelines for optimizing MLLMs, enabling a balance between model size, performance, and data efficiency to meet specific deployment resource constraints. Due to computational constraints, in this work we focused on two representative pruning techniques on two different models. Future work could extend these best practices to include a broader range of pruning techniques and models to further refine these strategies.

## REPRODUCIBILITY STATEMENT

We have made extensive efforts to ensure the reproducibility of our results. All pruning and recovery strategies are thoroughly detailed in Section 3. Section 4 provides a comprehensive overview of the experimental setup, with additional details included in the Supplementary Material. Upon publication, we will release the source code for pruning, recovery training, and the compressed model checkpoints, allowing researchers to replicate and extend our work. For evaluation, we used a widely adopted evaluation suite, which is publicly available and cited in our paper, ensuring a consistent and transparent benchmarking process.

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

# A    MODEL ARCHITECTURE OF LLaVA AND BUNNY

Table 4 outlines the architectures of the Bunny and LLaVA models. LLaVA-v1.5-7B employs CLIP-ViT-L (Radford et al., 2021) as the vision encoder and Vicuna-v1.5 Chiang et al., 2023 as the language decoder, while Bunny-v1.0-3B utilizes SigLIP-SO (Zhai et al., 2023) as the vision encoder and Phi-2 (Javaheripi et al., 2023) as the language decoder. Both models leverage MLP layers to align the vision and language modalities.

| Model | Parameters | Vision Encoder | Multimodal Projector | Language Decoder |
|---|---|---|---|---|
| LLaVA-v1.5-7B | 7.0B | CLIP-ViT-L (0.3B) | mlp2x-gelu (0.01B) | Vicuna-v1.5 (6.7B) |
| Bunny-v1.0-3B | 3.2B | SigLIP-SO (0.4B) | mlp2x-gelu (0.02B) | Phi-2 (2.8B) |

Table 4: Architecture details of the uncompressed models. We present the number of parameters, along with the vision encoder, multimodal projector and the language decoder of the models included in our study.

# B    IMPLEMENTATION DETAILS OF HIDDEN STATE MATCHING

To determine which LLM layers' hidden states to map between the pruned and unpruned models, we explore three options: matching the last layer, the last two, and the last three layers. Table 5 shows that matching only the last layer's hidden state yields the best performance.

| Ratio | Layer-1 | Layer-1,2 | Layer-1,2,3 |
|---|---|---|---|
| 12.8% | 95.34% | 95.17% | 96.25% |
| 25.5% | 91.02% | 90.48% | 90.97% |
| 39.0% | 87.08% | 86.12% | 84.84% |
| 51.8% | 75.25% | 72.56% | 72.87% |

Table 5: Results for recovering widthwise pruned Bunny with hidden state mapping. We compare the relative performance for mapping the last layer (layer-1), the last two layers (layer-1,2), and the last three layers (layer-1,2,3). By only mapping the last LLM layer the best performance is achieved.

# C    MORE RESULTS FOR PRUNING

Table 6 presents the model performance for widthwise and layerwise pruning. For both of the Bunny and LLaVA models, widthwise pruning consistently outperforms layerwise pruning. This performance gap widens as the compression ratio increases, with widthwise pruning showing a more significant advantage at higher compression ratios.

# D    MORE RESULTS ON MODEL EFFICIENCY

For the models pruned by layerwise method, we also assess their memory consumption as FLOPs. Memory consumption refers to the allocated GPU memory, while FLOPS are measured using the Calflops codebase[2]. The results in Table 7 show the same trend as widthwise pruning, indicating that the achieved compressions directly translate into improvements in memory efficiency and computational cost for both widthwise and layerwise pruning.

# E    KNOWLEDGE DISTILLATION WITH AND WITHOUT FINETUNING

Figure 5 and Figure 6 compare logits-based knowledge distillation (represented by RKL) and hidden state matching-based knowledge distillation (represented by L2 loss), with and without supervised fine-tuning, following widthwise and layerwise pruning, respectively. While knowledge distillation

---

[2]Caflops codebase: https://github.com/MrYxJ/calculate-flops.pytorch

| Method | Size | PruneRatio | MMMU | GQA | SQA | MME-C | MME-P | POPE | AVG | AVG-% |
|---|---|---|---|---|---|---|---|---|---|---|
| LLaVA-v1.5-7B | 7.0B | | 35.10 | 61.98 | 68.67 | 363.21 | 1511.33 | 86.99 | 62.28 | 100.00% |
| Width-wise | 6.3B | 15% | 32.40 | 59.34 | 63.21 | 268.93 | 1432.47 | 86.57 | 57.79 | 92.79% |
| | 5.5B | 30% | 31.00 | 52.59 | 54.29 | 253.21 | 1174.93 | 86.29 | 52.43 | 84.17% |
| | 4.8B | 45% | 27.60 | 20.86 | 12.10 | 70.00 | 347.45 | 45.96 | 22.11 | 35.49% |
| | 4.0B | 60% | 23.30 | 0.43 | 0.40 | 2.14 | 19.24 | 3.94 | 4.88 | 7.84% |
| Depth-wise | 6.3B | 15% | 31.80 | 42.77 | 55.23 | 202.14 | 701.83 | 86.38 | 46.09 | 74.00% |
| | 5.5B | 30% | 32.70 | 42.18 | 59.64 | 210.71 | 921.88 | 78.69 | 47.61 | 76.43% |
| | 4.8B | 45% | 26.90 | 14.39 | 3.82 | 132.86 | 616.63 | 51.69 | 24.04 | 38.60% |
| | 4.0B | 60% | 25.80 | 0.00 | 0.00 | 0.00 | 0.00 | 0.00 | 4.30 | 6.90% |
| Bunny-v1_0-3B | 3.2B | | 34.10 | 54.72 | 70.70 | 289.30 | 1487.71 | 87.82 | 59.65 | 100.00% |
| Width-wise | 2.8B | 15% | 30.90 | 51.83 | 65.64 | 242.50 | 1207.85 | 87.94 | 54.50 | 95.48% |
| | 2.5B | 30% | 28.40 | 45.65 | 55.73 | 199.64 | 807.95 | 87.13 | 47.04 | 87.57% |
| | 2.0B | 45% | 25.70 | 37.92 | 3.42 | 200.00 | 618.25 | 83.12 | 34.35 | 60.66% |
| | 1.6B | 60% | 24.80 | 6.12 | 0.00 | 141.07 | 293.23 | 2.34 | 10.93 | 13.52% |
| Depth-wise | 2.8B | 15% | 33.80 | 29.42 | 69.66 | 271.43 | 1456.41 | 87.91 | 54.59 | 91.52% |
| | 2.5B | 30% | 29.00 | 24.77 | 28.76 | 272.86 | 1273.34 | 86.50 | 44.47 | 74.55% |
| | 2.0B | 45% | 23.90 | 16.85 | 3.47 | 191.43 | 867.37 | 80.09 | 31.94 | 53.54% |
| | 1.6B | 60% | 26.60 | 0.02 | 17.15 | 0.71 | 55.92 | 0.02 | 7.78 | 13.04% |

Table 6: Pruning results for LLaVA-v1.5-7B and Bunny-v1-3B. Size is the number of total parameters of the model, while the compression ratio (Ratio) indicates the proportion of remaining language model parameters compared to the pre-pruning state. For both models, width-wise pruning results in better performance without finetuning compared to depth-wise pruning.

| Ratio | Bunny | | LLaVA | |
|---|---|---|---|---|
| | Mem. (MiB) | FLOPS (T) | Mem. (MiB) | FLOPS (T) |
| 0% | 6,167 | 4.77 | 13,546 | 9.57 |
| 15% | 5,411 | 4.16 | 11,604 | 8.03 |
| 30% | 4,659 | 3.56 | 9,664 | 6.92 |
| 45% | 3,907 | 2.95 | 7,724 | 5.55 |
| 60% | 3,006 | 2.22 | 5,496 | 3.9 |

Table 7: Memory requirements (Mem.) and FLOPS for the Bunny and LLaVA models at various compression ratios. The models are pruned layerwise. The evaluation is performed in inference mode, where each model is provided with an image and a prompt containing 50 tokens.

alone helps in recovering performance post-pruning, it remains less effective than supervised fine-tuning. However, when combined with supervised fine-tuning, it results in superior performance.

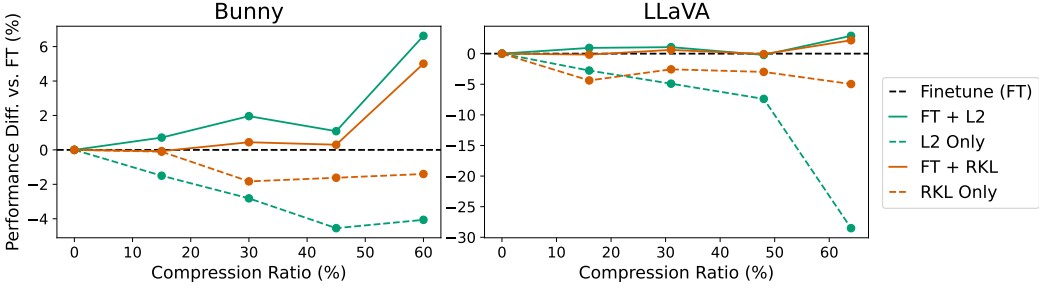

Figure 5: Comparison of L2 and RKL distillation strategies with and without additional fine-tuning loss for Bunny and Llava models compressed by widthwise pruning. The plot shows performance differences relative to standard fine-tuning across varying compression ratios.

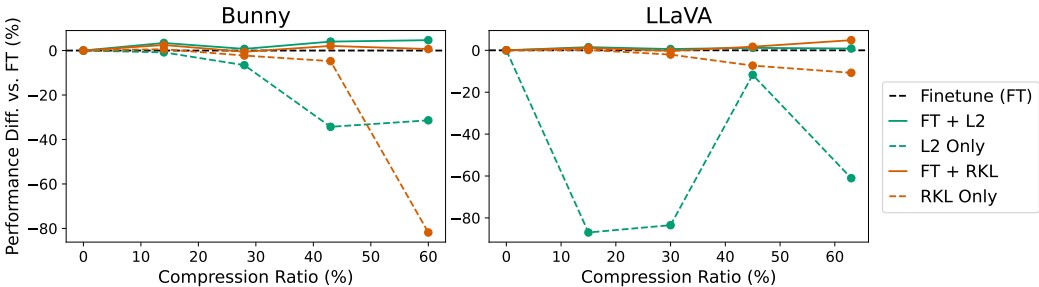

Figure 6: Comparison of L2 and RKL distillation strategies with and without additional fine-tuning loss for Bunny and Llava models compressed by layerwise pruning. The plot shows performance differences relative to standard fine-tuning across varying compression ratios.

## F KNOWLEDGE DISTILLATION STRATEGIES COMPARISON AFTER LAYERWISE PRUNING

Figure 7 compares various distillation strategies based on their relative improvement over finetuning alone after layerwise pruning. Similar to the results after widthwise pruning, applying hidden states matching yields the best performance, or at least matches other methods. The trend that RKL generally outperforms KL is also observed here.

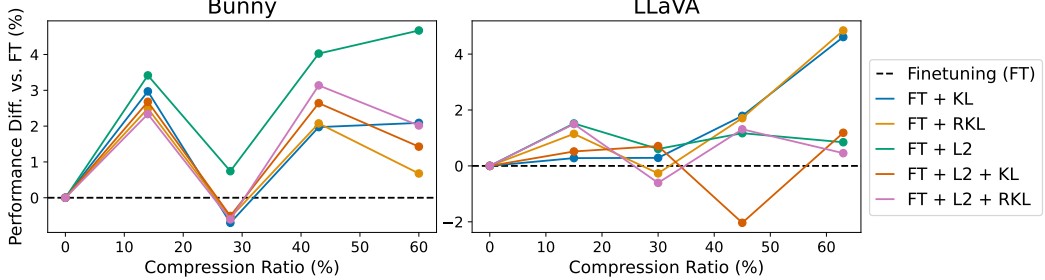

Figure 7: Comparison of different distillation recovery strategies (KL loss, RKL loss, L2 loss, and their combinations) for Bunny and LlaVA models pruned with layerwise pruning. The plot shows the relative performance improvement of each strategy over standard fine-tuning across various compression ratios. The results demonstrate that distillation helps recover more performance than fine-tuning alone, with the L2 loss component consistently leading to the largest performance gains, particularly at higher compression ratios.

## G QUALITATIVE RESULTS OF THE COMPRESSED MODELS

Table 8 shows the qualitative results of the compressed LLaVA models. After compression, the models demonstrate a strong ability to comprehend visual inputs and generate detailed textual outputs.

## H PRUNING VS. QUANTIZATION

The table includes results for LLaVA-6B and LLaVA-5B, where the former is layerwise pruned and recovery trained, and the latter is widthwise pruned and recovery trained. Our findings indicate that combining structured pruning with quantization provides a balanced approach: it significantly reduces memory usage while mitigating the latency overhead associated with quantization alone. This synergy leverages the strengths of both methods, offering an effective strategy for compressing large models while maintaining performance and efficiency.

| Model | Response |
|---|---|
| |  |
| | **Prompt**: A chat between a curious user and an artificial intelligence assistant. The assistant gives helpful, detailed, and polite answers to the user's questions. USER: $<image>$ What is funny in the image? |
| LLaVA-6.3B | The image is funny because it features a dog wearing a Renaissance-style dress and a bandana, posing as if it's posing for a painting. This is humorous because it is a playful and unconventional representation of a dog, which is not something you would typically see in a painting or a Renaissance-style dress. The dog's pose and attire create a sense of humor and absurdity, making the image entertaining and amusing to viewers. |
| LLaVA-5.5B | The image is funny because it features a dog dressed up in a human-like attire, wearing a long black dress and a hat. This unusual and amusing scene is not something you would typically see in real life, as dogs don't wear clothes or accessories like humans. The dog's expression and the way it is posing for the picture further contribute to the comic effect of the image. The image captures the viewer's attention and sparks a sense of humor due to the unexpected and imaginative representation of a dog dressed like a human. |
| LLaVA-3.8B | The image is funny because it features a dog dressed in a human-like dress, posing in a pose as if it were a person. The dog appears to be dressed in a dress, which is a humorous and unexpected scene, as dogs are not typically dressed like humans. The dog's pose and the overall scene create a sense of humor and playfulness, making the image a delightful and entertaining piece. |

Table 8: Qualitative analysis of compressed LLaVA models with respect to the image. We feed the compressed LLaVA models the image paired with a prompt to generate a response. The compressed LLaVA models are able to understand the visual inputs and output reasonable texts.

## I  GENERALIZABILITY OF THE BEST PRACTICES

We also explored the impact of recovery training, specifically finetuning the multimodal projector and the language model, as well as the importance of incorporating knowledge distillation. These results, shown in Table 11, confirm that finetuning only the multimodal projector is sufficient at small compression ratios, as pruning minimally impacts the language model but disrupts multimodal alignment. For InternVL, finetuning the projector alone recovers 96.9% of the performance at a 15% compression ratio, compared to 97.8% when both the projector and the language model are finetuned. At a 30% compression ratio, projector-only finetuning recovers 75.1% while finetuning both components recovers 86.6%. Furthermore, combining supervised finetuning with intermediate

| Model | Memory(MiB) | Ratio | MMMU | GQA | SQA | MME-C | MME-P | POPE | AVG | Latency |
|---|---|---|---|---|---|---|---|---|---|---|
| LLaVA-7B | 13546 | | 35.10 | 61.98 | 68.67 | 363.21 | 1511.33 | 86.99 | 62.28 | 105ms ± 1.5ms |
| LLaVA–7B.int8() | 7518 | | 35.2 | 61.87 | 68.22 | 350.71 | 1508.41 | 86.54 | 61.85 | 398ms ± 1.31ms |
| LLaVA-6B | 11604 | 15% | 35.40 | 61.17 | 68.07 | 328.57 | 1454.20 | 86.51 | 60.82 | 95 ms ± 8.1 ms |
| LLaVA-6B.int8() | 6473 | 15% | 35.40 | 61.17 | 68.07 | 328.57 | 1454.20 | 86.51 | 60.82 | 125 ms ± 937 $\mu s$ |
| LLaVA-5B | 9548 | 30% | 31.80 | 60.71 | 60.54 | 252.50 | 1407.08 | 86.68 | 56.94 | 80.7 ms ± 634 $\mu s$ |
| LLaVA-5B.int8() | 5389 | 30% | 31.6 | 60.65 | 60.09 | 263.57 | 1410.28 | 86.78 | 57.10 | 141 ms ± 2.4 ms |

Table 9: Pruning and Quantization comparison on LLaVA-v1.5-7B.

| Model | Size | Ratio | MMMU | GQA | SQA | MME-C | MME-P | POPE | AVG | AVG-% |
|---|---|---|---|---|---|---|---|---|---|---|
| InternVL-Chat-4B-V1-5 | 4B | | 43.20 | 62.57 | 93.30 | 547.50 | 1,596.71 | 88.00 | 72.56 | 100% |
| Layerwise | 3.5B | 15% | 42.70 | 54.43 | 92.96 | 527.86 | 1,534.37 | 88.09 | 70.15 | 96.68% |
| Widthwise | 3.5B | 15% | 43.60 | 56.35 | 93.12 | 510.10 | 1,588.30 | 87.96 | 70.70 | 97.44% |

Table 10: Comparison of different pruning techniques on the model InternVL-Chat-4B.

representation distillation consistently achieves the highest performance across all compression ratios. For InternVL, this approach recovers 98.2% of performance at 15% compression and 87.2% at 30%, underscoring its effectiveness.

Overall, the results from InternVL demonstrate that our best practices generalize effectively to different model architectures, confirming their applicability and robustness.

| Model | Size | Ratio | MMMU | GQA | SQA | MME-C | MME-P | POPE | AVG | AVG-% |
|---|---|---|---|---|---|---|---|---|---|---|
| Mini-InternVL-Chat-4B-V1-5 | 4B | - | 43.20 | 62.57 | 93.30 | 547.50 | 1,596.71 | 88.00 | 72.56 | 100% |
| SFT mm only | 3.5B | 15% | 42.70 | 54.43 | 92.96 | 527.86 | 1,534.37 | 88.09 | 70.15 | 96.68% |
| SFT mm + LLM | 3.5B | 15% | 43.10 | 56.34 | 93.36 | 524.64 | 1,585.83 | 88.10 | 70.96 | 97.80% |
| SFT+KD mm + LLM | 3.5B | 15% | 43.30 | 56.17 | 93.41 | 539.64 | 1,582.58 | 87.89 | 71.23 | 98.16% |
| SFT mm only | 3B | 30% | 33.30 | 27.39 | 62.82 | 197.14 | 845.60 | 73.20 | 43.94 | 60.56% |
| SFT mm + LLM | 3B | 30% | 34.40 | 53.46 | 76.80 | 432.27 | 1,438.28 | 86.57 | 62.86 | 86.64% |
| SFT+KD mm + LLM | 3B | 30% | 36.20 | 53.77 | 76.60 | 448.21 | 1,410.88 | 86.60 | 63.29 | 87.23% |

Table 11: Comparison of recovery training only multimodal projector (mm) and large language model on Mini-InternVL-Chat-4B-V1-5 after layerwise pruning with different recovery strategies, i.e., supervised finetuning (SFT) and knowledge distillation (KD) on intermediate representations.

