# OpenReview forum: "From Bulk to Budget: Best Practices To Compress Multimodal Large Language Models"
_ICLR.cc/2025/Conference — Submitted to ICLR 2025_

### Official Review · Reviewer_zH8E · 2024-10-31

**Soundness:** 2
**Presentation:** 3
**Contribution:** 2
**Rating:** 5
**Confidence:** 4

**Summary:**

This paper provides a recipe to compress MLLM by evaluating the width-wise pruning and layerwise pruning, finding that widthwise pruning is better. Then, by recovery strategies like SFT and KD to restore the performance.

**Strengths:**

- This paper is easy to understand and follow.
- This paper provides concrete best practices for practitioners

**Weaknesses:**

- The experiments are not comprehensive with just limited model selection. Please include newer MLLM architectures like InternVL, CogVLM, MiniCPM-v;
- The title of this paper claimed "Best Practices To Compress MLLM". However, this paper only focuses on pruning and then knowledge distillation. Additionally, the pruning only focuses on LLM and there is no experiment on pruning Vision encoder. It is a little bit overclaimed when you do not involve other compression methods like quantization or low-rank factorization. This paper should dive into compression and provide more detailed insight for readers.
- Limited novelty: pruning+knowledge distillation are the common techniques on LLM, as presented in Sheared LLaMA and Minitron. This paper has limited novelty as there is no difference.
- Limited comparison with other compression methods: this paper proposed several techniques like width/depth pruning and KD. This paper did not involve any compression method with other methods.

**Questions:**

See Weakness

---

> ### Author Response · Authors · 2024-11-22
> **Author Response to Reviewer zH8E**
>
> >The experiments are not comprehensive with just limited model selection. Please include newer MLLM architectures like InternVL, CogVLM, MiniCPM-v;
>
> We agree it benefits our study to evaluate our methods on a broader range of MLLM architectures. In response, we have extended our experiments to include the InternVL [1] model to test the generalizability of our best practices for pruning and recovery. The results are in the revision Appendix I. Our findings with InternVL confirm that the proposed methods are effective across architectures, demonstrating the applicability of our techniques beyond the models initially presented.
>
> We provide an analysis of the results of the new model below.
> We observe that widthwise pruning offers better performance without recovery training. With InternVL, widthwise pruning retains 97.4% of the model’s original performance at a 15% compression ratio, compared to 96.7% for layerwise pruning, reinforcing its suitability as a default strategy in low-resource scenarios.
>
> | Model | Size | Ratio | MMMU  | GQA   | SQA   | MME-C | MME-P | POPE | AVG   | AVG-% |
> |-------------------------------------|------|-------|-------|-------|-------|-------|-------|------|-------|--------|
> | Mini-InternVL-Chat-4B-V1-5 | 4B   | | 43.20 | 62.57 | 93.30 | 547.50 | 1,596.71 | 88.00 | 72.56 | 100%   |
> | Layerwise (Prune Only)| 3.5B | 15%   | 42.70 | 54.43 | 92.96 | 527.86 | 1,534.37 | 88.09 | 70.15 | 96.68% |
> | Widthwise (Prune Only)| 3.5B | 15%   | 43.60 | 56.35 | 93.12 | 510.10 | 1,588.30 | 87.96 | 70.70 | 97.44% |
>
> Additionally, we find that finetuning only the multimodal projector is sufficient at small compression ratios, where pruning minimally impacts the language model but disrupts multimodal alignment. These results reinforce our observations from earlier models and validate the transferability of our proposed practices.
> | Model| Size | Ratio | MMMU | GQA   | SQA   | MME-C | MME-P | POPE | AVG   | AVG-% |
> |-------------------------------------|------|-------|-------|-------|-------|-------|-------|------|-------|--------|
> | Mini-InternVL-Chat-4B-V1-5  | 4B| |43.20 | 62.57 | 93.30 | 547.50 | 1,596.71 | 88.00 | 72.56 | 100%   |
> | Layerwise Prune + Finetuning mm-projector | 3.5B | 15%   | 42.70 | 54.43 | 92.96 | 527.86 | 1,534.37 | 88.09 | 70.15 | 96.68% |
> | Layerwise Prune + Finetuning mm-projector | 3B   | 30%   | 33.30 | 27.39 | 62.82 | 197.14 | 845.60  | 73.20 | 43.94 | 60.56% |
>
> | Model |Size| Ratio| MMMU| GQA| SQA| MME-C| MME-P|POPE| AVG| AVG-% |
> |-------------------------------------|------|-------|-------|-------|-------|-------|-------|------|-------|--------|
> | Mini-InternVL-Chat-4B-V1-5| 4B| |43.20 | 62.57 | 93.30 | 547.50 | 1,596.71 | 88.00 | 72.56 | 100%   |
> | Layerwise Prune + Finetuning mm & LLM|3.5B| 15%   | 43.10 | 56.34 | 93.36 | 524.64 | 1,585.83 | 88.10 | 70.96 | 97.80% |
> | Layerwise Prune + Finetuning mm & LLM| 3B   | 30%   | 34.40 | 53.46 | 76.80 | 432.27 | 1,438.28 | 86.57 | 62.86 | 86.64% |
>
> Moreover, our initial findings show that combining supervised finetuning with intermediate representation distillation consistently yields the highest performance across compression ratios. With InternVL, this combined approach achieves 98.2% recovery at a 15% compression ratio and 87.2% at a 30% compression ratio, confirming its effectiveness.
>
> | Model | Size | Ratio | MMMU| GQA| SQA|MME-C|MME-P|POPE| AVG| AVG-%|
> |-------------------------------------|------|-------|-------|-------|-------|-------|-------|------|-------|--------|
> | Mini-InternVL-Chat-4B-V1-5 | 4B| 43.20 | 62.57 | 93.30 | 547.50 | 1,596.71 | 88.00 | 72.56 | 100%   |
> | Layerwise Prune + Finetuning + Distillation | 3.5B | 15% | 43.30 | 56.17 | 93.41 | 539.64 | 1,582.58 | 87.89 | 71.23 | 98.16% |
> | Layerwise Prune + Finetuning + Distillation | 3B   | 30% | 36.20 | 53.77 | 76.60 | 448.21 | 1,410.88 | 86.60 | 63.29 | 87.23% |
> | Layerwise Prune + Finetuning + Distillation | 2.5B | 45% | 35.00 | 44.22 | 37.08 | 142.50 | 991.94  | 81.80 | 44.25 | 60.99% |
>
> Notably, All of the pruned models are recovery trained on only 3% of the original dataset, which highlights the data efficiently. Overall, the results from InternVL indicate that our methods generalize well to newer MLLM architectures. We will incorporate these findings in the final version of the paper to strengthen our contributions.

---

> > ### Author Response · Authors · 2024-11-22
> > **Author Response to Reviewer zH8E**
> >
> > >  * The title of this paper claimed "Best Practices To Compress MLLM". However, this paper only focuses on pruning and then knowledge distillation. Additionally, the pruning only focuses on LLM and there is no experiment on pruning Vision encoder. It is a little bit overclaimed when you do not involve other compression methods like quantization or low-rank factorization. This paper should dive into compression and provide more detailed insight for readers.
> > >  * Limited comparison with other compression methods: this paper proposed several techniques like width/depth pruning and KD. This paper did not involve any compression method with other methods.
> >
> > We thank the reviewer for their detailed feedback. In the majority of the existing MLLMs, the parameter count is dominated by the LLM. Compressing the vision encoder has a comparatively smaller impact on the overall model size. Hence, our main contribution lies in systematically adapting, comparing, and analyzing existing LLM compression techniques to MLLMs. We identify and address the unique challenges posed by their multimodal nature.
> >
> > We agree that incorporating additional compression methods broadens the scope of our work. To address this, we have conducted experiments with 8-bit quantization applied to both the LLaVA base model and the pruned model with recovery training and summarize the results in the revision Appendix H. Our results show that quantization reduces the memory footprint of the base model by 44.5%, with only a 0.43% drop in average performance. However, we observe a fourfold increase in model latency.
> >
> > | Model                               | Memory  | Ratio | MMMU | GQA   | SQA   | MME-C  | MME-P  | POPE  | AVG   | Latency                |
> > |-------------------------------------|---------|-------|------|-------|-------|--------|--------|-------|-------|------------------------|
> > | LLaVA-v1.5-7B                       | 13546MiB|       | 35.10| 61.98 | 68.67 | 363.21 | 1511.33 | 86.99 | 62.28 | 105ms ± 1.5ms          |
> > | LLaVA-v1.5-7B.int8()                | 7518MiB |       | 35.2 | 61.87 | 68.22 | 350.71 | 1508.41 | 86.54 | 61.85 | 398ms ± 1.31ms         |
> > | LLaVA-6B-layerwise+recovery         | 11604MiB| 15%   | 35.40| 61.17 | 68.07 | 328.57 | 1454.20 | 86.51 | 60.82 | 95ms ± 8.1ms           |
> > | LLaVA-6B-layerwise+recovery.int8()  | 6473MiB | 15%   | 35.40| 61.17 | 68.07 | 328.57 | 1454.20 | 86.51 | 60.82 | 125ms ± 937μs          |
> > | LLaVA-5B-layerwise+recovery         | 9548MiB | 30%   | 31.80| 60.71 | 60.54 | 252.50 | 1407.08 | 86.68 | 56.94 | 80.7ms ± 639μs         |
> > | LLaVA-5B-layerwise+recovery.int8()  | 5389MiB | 30%   | 31.6 | 60.65 | 60.09 | 263.57 | 1410.28 | 86.78 | 57.10 | 141ms ± 2.4ms          |
> >
> > We also tested the combination of quantization with layerwise pruning and recovery training. As shown in the above table,  at 15% and 30% compression ratios, the memory footprint was reduced by 40% and 44%, respectively, with only 0.4% and 1.3%  reductions in average performance. These results highlight the complementarity of structured pruning and quantization as effective compression techniques.
> >
> >
> > >Limited novelty: pruning+knowledge distillation are the common techniques on LLM, as presented in Sheared LLaMA and Minitron. This paper has limited novelty as there is no difference.
> >
> >
> > We want to clarify that our paper is intended as a best practices study rather than a proposal of a novel compression method, which we do not claim in our contributions. Our primary contribution lies in systematically adapting, comparing, and analyzing existing compression techniques to MLLMs. We evaluate various structured pruning strategies and recovery methods, quantifying the trade-offs between compression ratio, performance retention, and data efficiency. By establishing these best practices, we provide practitioners with actionable insights and practical guidance for selecting appropriate techniques for MLLM compression based on specific deployment requirements.
> >
> > Additionally, while Sheared LLaMA and Minitron focus on LLM compression, our work targets MLLMs and their unique challenges. One such challenge is the issue of modality misalignment caused by compression, which we address in Section 4.2. Our findings demonstrate that fine-tuning the multimodal projector alone can recover over 95% of the performance at lower compression ratios, as this step realigns textual and visual features. For higher compression ratios, we show that fine-tuning both the projector and the language model further mitigates performance degradation. These contributions address challenges unique to MLLMs, which are not explored in Sheared LLaMA or Minitron.
> >
> >
> > Reference
> >
> > [1] Chen et al. Internvl: Scaling up vision foundation models and aligning for generic visual-linguistic tasks. CVPR 2024.

---

> > > ### Comment · Reviewer_zH8E · 2024-11-24
> > > **Response to authors.**
> > >
> > > We thank the authors for the additional experiments on InternVL and quantization, which partially address some concerns. However, the main issue of overclaiming remains unresolved. The paper’s title suggests a comprehensive exploration of MLLM compression, yet it remains narrowly focused on pruning and knowledge distillation, with limited novelty and insufficient comparisons to other compression methods.
> > >
> > > The paper would benefit from a broader scope and deeper insights to align with its claims. Thus, I maintain my rating and encourage the authors to refactor the paper for greater clarity and contribution.

---

> > > > ### Author Response · Authors · 2024-11-26
> > > > **Author Response to Reviewer zH8E**
> > > >
> > > > Dear Reviewer zH8E,
> > > >
> > > > The deadline for submitting the revised PDF is approaching. If you have any additional suggestions or concerns you'd like us to address in the revised paper, please let us know, and we’ll gladly incorporate them as soon as possible before the deadline. Thank you!

---

> ### Author Response · Authors · 2024-11-24
> **Author Response to Reviewer  zH8E**
>
> We agree with the reviewer on the scope of the paper and would be happy to change the title to “From Bulk to Budget: Best Practices To Compress Multimodal Large Language Models through Structural Pruning and Recovery.” and refactor our paper accordingly to clarify the contribution. Nevertheless we would like to emphasize that the contributions of our paper —offering best practices for MLLM compression via pruning and recovery—are substantial. We believe these insights will be valuable to a broader audience and practitioners.

---

> > ### Comment · Reviewer_zH8E · 2024-11-27
> >
> > Thanks for the rebuttal. Even if you plan to change the title of this paper, there are still several problems.
> >
> > Although you claim that this paper is a benchmark not a novel algorithm, it still requires other pruning methods like SparseGPT, Wanda, GBLM pruner etc.
> >
> > My suggestion is:
> >
> > 1. revise the title (even for pruning itself, this paper does not deserve the 'best practice')
> > 2. add more pruning methods
> > 3. add sota VLMs
> >
> > Different from LLM, pruning on VLMs should have some distinct characters. Please reveal more insights in your paper.
> > If you can make the above changes, I think this paper can reach the level of acceptance.

---

> > > ### Author Response · Authors · 2024-11-28
> > > **Author Response to Reviewer zH8E**
> > >
> > > We thank Reviewer zH8E for the very constructive suggestions.
> > >
> > > > Revise the title (even for pruning itself, this paper does not deserve the 'best practice')
> > >
> > > We agree with the reviewer on the scope of the paper and would be happy to change the title to "From Bulk to Budget: Structural Pruning Multimodal Large Language Model in Practice."
> > >
> > > > Add more pruning methods
> > >
> > > The pruning methods, including SparseGPT, Wanda, and GBLM, require a much larger calibration dataset (128 examples). In MLLMs, including image tokens results in much longer input sequences. For example, for the Bunny model, which uses "siglip-so400m-patch14-384" as an encoder, the resulting image tokens have a length of 729. This results in higher memory and computational demands for importance estimation compared to the layerwise and widthwise pruning methods used in our paper, where we effectively rely on just 10 examples for both methods. However, we appreciate the suggestion and will incorporate more pruning methods in future work.
> > >
> > > > Add sota VLMs
> > >
> > > We extend our study to include InternVL, one of the state-of-the-art VLMs, and are pleased to align our practices with current advancements. Moving forward, we will continue to update our work by including more VLMs to ensure its relevance and comprehensiveness.

---

> > > ### Author Response · Authors · 2024-11-30
> > > **Author Response to Reviewer zH8E**
> > >
> > > Dear Reviewer zH8E,
> > >
> > > We sincerely thank you for your thoughtful and constructive feedback. Your insights have significantly contributed to the refinement of our paper, particularly regarding experiment design, scope, and the insights provided.
> > >
> > > In response, we conducted additional experiments to combine quantization with pruning, demonstrating the complementarity of these techniques as effective methods for model compression. Moreover, we extended our best practices to the InternVL model, showcasing the generalizability of our approach to newer MLLM architectures. Additionally, we revised our experiment session to provide insights on MLLM-specific features. For example, in Section 4.2, we investigate how compression affects the alignment of textual and visual features, which is essential for MLLM performance. Lastly, we're happy to revise the title and refactor the paper for greater clarity and contribution.
> > >
> > > Our paper aims to offer actionable insights and practical techniques for MLLM compression through pruning and knowledge distillation, helping practitioners save time and computational resources. We are also committed to updating the paper with state-of-the-art methods and models.
> > >
> > > In light of these enhancements, we kindly ask you to reconsider and potentially adjust your review score, considering the improvements and additional evidence provided during the rebuttal. Your feedback has been invaluable in improving our work, and we deeply appreciate your time and effort. Thank you again for your thoughtful review and for considering our request.

---

### Official Review · Reviewer_J6JH · 2024-11-02

**Soundness:** 2
**Presentation:** 3
**Contribution:** 2
**Rating:** 3
**Confidence:** 5

**Summary:**

This paper presents a comprehensive study on compressing Multimodal Large Language Models (MLLMs) while maintaining performance. The authors investigate two pruning strategies (widthwise and layerwise) and various recovery methods (supervised finetuning and knowledge distillation) across different compression ratios. They conduct experiments on two MLLMs (LLaVA-v1.5-7B and Bunny-v1.0-3B) and provide best practices for MLLM compression based on their findings.

**Strengths:**

The paper provides extensive empirical evaluations across different compression ratios, recovery strategies, and data requirements.

The experiments are well-designed and cover multiple dimensions: pruning methods, and data efficiency.

**Weaknesses:**

1. The authors' claim of being the first to investigate general-purpose MLLM compression overlooks existing work, particularly the survey paper on efficient MLLMs [1], which contradicts the authors’ claim. The literature review could be more comprehensive, especially in acknowledging related work in MLLM efficiency.

2. The technical contributions largely adapt existing LLM compression techniques to MLLMs without introducing significant novel methods.

3. The findings mostly confirm expected behaviors from LLM compression research. The paper primarily combines existing techniques rather than introducing new methodological advances. The exploration could better highlight MLLM-specific challenges and solutions that differentiate it from general LLM compression


[1] Efficient Multimodal Large Language Models: A Survey

**Questions:**

see weaknesses

---

> ### Author Response · Authors · 2024-11-22
> **Author Response to Reviewer J6JH**
>
> >The authors' claim of being the first to investigate general-purpose MLLM compression overlooks existing work, particularly the survey paper on efficient MLLMs [1], which contradicts the authors’ claim. The literature review could be more comprehensive, especially in acknowledging related work in MLLM efficiency.
>
> We thank the reviewer for pointing out this related work in MLLM efficiency. We recognize the value of the survey; however, our focus is different. The survey investigates building efficient small MLLMs using pretrained efficient components and techniques rather than compressing existing MLLMs to meet specific size requirements. The survey offers an excellent overview of the existing literature on efficient MLLMs while we conducted extensive experiments, i.e. combination of two pruning methods, two ways of finetuning, three different knowledge distillation losses across two models, to find the best practices for compressing MLLMs. This experimental rigor distinguishes our contribution from the broader scope of the survey.
>
> We recognize the value of efficient MLLMs mentioned in the survey. While building MLLMs based on the pretrained efficient components such as SLMs and optimized vision models does help reduce the model size and improve the latency, the fixed size of the underlying components constrains their flexibility. Furthermore, training an efficient component such as SLM [2] from scratch to meet desired specifications is computationally expensive.
>
> Our work explicitly targets methods for customizing the size of existing MLLMs through structured pruning and recovery strategies. Notably, recovery training only uses a tiny fraction of the training data, making it more cost-efficient than training a smaller MLLM or SLM from scratch.
>
> We have included [1] and other relevant studies on efficient MLLMs in the related work (Sec. 2) in our revision.

---

> ### Author Response · Authors · 2024-11-22
> **Author Response to Reviewer J6JH**
>
> >The technical contributions largely adapt existing LLM compression techniques to MLLMs without introducing significant novel methods.
>
> Indeed, we are not presenting a new compression method. However, we emphasize the main contribution of this paper is best practices for MLLM compression. To our knowledge, this is the first systematic investigation of pruning and recovery training techniques on MLLMs, as pointed out by reviewer T1Q7.
>
> In the majority of the existing MLLMs, the parameter count is dominated by the LLM. Hence, our main contribution lies in systematically adapting, comparing, and analyzing existing LLM compression techniques to MLLMs. We identify and address the unique challenges posed by their multimodal nature.
>
> We evaluate various structured pruning strategies and recovery methods, quantifying the trade-offs between compression ratio, performance retention, and data efficiency. In the revision, we also added 8-bit quantization experiments on the LLaVA model. The results are in the updated Appendix H.  There, we demonstrate that pruning and recovery training can be effectively combined with quantization to achieve minimal performance loss.
>
> Furthermore, we extend our best practices to the new model Mini-InternVL-Chat-4B-V1-5 [1].  The results are in the updated Appendix I. Our additional experiments with InternVL confirm that our findings generalize and validate our best practices transfer to other MLLM architectures.
> In summary, by establishing these best practices, we provide practitioners with insights and practical guidance for selecting appropriate techniques for MLLM compression based on specific deployment needs, which we believe is a strong technical contribution.
>
> >The findings mostly confirm expected behaviors from LLM compression research. The paper primarily combines existing techniques rather than introducing new methodological advances. The exploration could better highlight MLLM-specific challenges and solutions that differentiate it from general LLM compression.
>
> We agree that certain observations from LLM compression carry over to the MLLM setting, such as the model retaining more performance when the compression ratio is smaller[3][4]. However, our work also addresses unique challenges specific to MLLMs. For example, in Section 4.2, we investigate how compression affects the alignment of textual and visual features, which is essential for MLLM performance. Our findings indicate that, at compression ratios up to 15%, fine-tuning only the projector network effectively recovers more than 95% of the model’s performance. This means that most degradation is due to misalignment between visual and textual feature space, thus suggesting a cost-efficient strategy for maintaining performance at lower compression levels. For larger compression ratios, we observe additional benefits from fine-tuning the language model as well.
>
> We appreciate the reviewer’s feedback and have emphasized the MLLM-specific aspects of our work more clearly in the revision section 4.2.
>
> Reference
>
> [1] Jin, Yizhang, et al. "Efficient multimodal large language models: A survey." arXiv 2024.
>
> [2]Chu, Xiangxiang, et al. "Mobilevlm: A fast, strong and open vision language assistant for mobile devices." arXiv 2023.
>
> [3] Ma et al. "Llm-pruner: On the structural pruning of large language models." NeurIPS 2023.
>
> [4] Men, Xin, et al. "Shortgpt: Layers in large language models are more redundant than you expect." arXiv 2024.

---

> ### Comment · Reviewer_J6JH · 2024-11-27
>
> The second concern I have about technical contribution is not addressed. The author even sold the concept of ''best practices for MLLM compression''; however, I do not accept it. Therefore, I keep my score as 3.

---

> > ### Author Response · Authors · 2024-11-27
> > **Author Response to Reviewer J6JH**
> >
> > >  The author even sold the concept of ''best practices for MLLM compression''
> >
> > Could you elaborate on what it means by this? The goal of the best practice paper is to provide practitioners with a comprehensive framework for MLLM compression, serving as a reference for those looking to compress their own MLLMs or a new MLLM in general. It consumes a lot of time and computing to try out all the pruning and KD methods and this paper aims to help practitioners save both time and resources. Additionally, the generalizability of these best practices to new models has been demonstrated, further supporting their utility.
> >
> > We genuinely want to know how to address your concern about technical contribution. Could you please clarify that?

---

> > ### Author Response · Authors · 2024-11-30
> > **Author Response to Reviewer J6JH**
> >
> > Dear Reviewer J6JH,
> >
> > Thank you for taking the time to review our work and for providing valuable feedback. The primary goal of our best-practice paper is to offer practitioners a comprehensive framework for MLLM compression. This serves as a practical reference for those seeking to compress their own MLLMs or new models in general. Given the significant time and computational resources required to experiment with various pruning and knowledge distillation methods, our work aims to streamline this process, enabling practitioners to save both time and resources effectively.
> >
> > Moreover, we have demonstrated the generalizability of these best practices to new models, underscoring their utility and relevance. We are also committed to maintaining the paper’s value over time by incorporating state-of-the-art methods and models in future updates.
> >
> > If there are any specific concerns or areas where you believe we could further clarify or enhance the work, we would be more than happy to discuss them and provide a detailed response. We would kindly ask you to reconsider your review in light of the additional evidence and improvements presented during the rebuttal phase.
> >
> > Thank you for your thoughtful consideration and support in improving the paper.

---

### Official Review · Reviewer_T1Q7 · 2024-11-03

**Soundness:** 3
**Presentation:** 3
**Contribution:** 2
**Rating:** 6
**Confidence:** 3

**Summary:**

This paper addresses the challenge of compressing multimodal large language models in a data-efficient way, introducing structured pruning and recovery techniques for resource-constrained deployments. The paper explores layerwise and widthwise pruning strategies, with a focus on identifying effective recovery techniques, such as finetuning and knowledge distillation. Key findings include the effectiveness of widthwise pruning in low-resource scenarios and the combination of projector finetuning with hidden state knowledge distillation for optimal performance recovery. The study provides insights into best practices for compressing MLLMs while maintaining high performance across a variety of tasks.

**Strengths:**

1. The paper presents a unique combination of widthwise and layerwise pruning for MLLMs, complemented by targeted recovery strategies like finetuning and knowledge distillation. This distinguishes the work from typical compression techniques that rely solely on one pruning or finetuning method, offering a more adaptable framework for diverse deployment needs.
2. This ppaer emphasizes data-efficient model recovery, highlighting scenarios where only 5% of the original data suffices to restore a substantial portion of the model’s performance, making the proposed method practical for environments where labeled data is scarce or costly, potentially expanding its applicability in low-data or real-time contexts.
3. Extensive experimentation across two MLLMs with varying compression ratios, recovery techniques, and data requirements provides a detailed view of how different configurations affect model performance. The ablation studies also offer a deeper understanding of the benefits and limitations of each pruning and recovery method.

**Weaknesses:**

1. The proposed techniques are evaluated within the context of MLLMs; however, they lack comparisons with other prevalent compression methods. This absence makes it challenging to assess their effectiveness against existing solutions, especially as structured pruning and knowledge distillation are already well-established in the field.
2. While the framework demonstrates potential at lower compression ratios, its performance gains are limited at higher compression ratios. This limitation may reduce the practical appeal of the method for applications that demand more aggressive compression while still preserving task performance.
3. Although the proposed combination of multiple pruning and recovery strategies is thorough, it increases implementation complexity. The reliance on complex data structures and recovery steps may hinder deployment. An author should clarify this.

**Questions:**

1. What are the limitations of the chosen pruning strategies for different types of tasks within MLLMs?
2. Can the proposed data-efficient recovery techniques be generalized to other model architectures or require specific adjustments?

---

> ### Author Response · Authors · 2024-11-22
> **Author Response to Reviewer T1Q7**
>
> >The proposed techniques are evaluated within the context of MLLMs; however, they lack comparisons with other prevalent compression methods. This absence makes it challenging to assess their effectiveness against existing solutions, especially as structured pruning and knowledge distillation are already well-established in the field.
>
> We appreciate the reviewer’s insight and understand the importance of comparing the performance of pruning and recovery strategies to other compression techniques. In particular, our findings highlight that quantization is a complementary compression approach that can be effectively combined with structured pruning, achieving significant memory savings with minimal performance loss.
>
> We included 8-bit quantization experiments on the LLaVA model and further analysis in the revision Appendix H to compare prevalent compression methods.
>
> These results demonstrate that quantization reduces the memory footprint of the base model by 44.5%, with only a 0.43pp decrease in average performance. However, quantization introduces a fourfold increase in model latency, which poses challenges for deployment scenarios requiring real-time inference. Additionally, we evaluated the combination of quantization with structured pruning and recovery training. For compression ratios of 15% and 30%, this combination reduced the memory footprint by 40% and 44%, respectively, while incurring only 0.4pp and 1.3pp drops in average performance. These experiments showcase the complementary nature of structured pruning and quantization as effective and practical compression strategies.
>
> | Model                               | Memory  | Ratio | MMMU | GQA   | SQA   | MME-C  | MME-P  | POPE  | AVG   | Latency                |
> |-------------------------------------|---------|-------|------|-------|-------|--------|--------|-------|-------|------------------------|
> | LLaVA-v1.5-7B                       | 13546MiB|       | 35.10| 61.98 | 68.67 | 363.21 | 1511.33 | 86.99 | 62.28 | 105ms ± 1.5ms          |
> | LLaVA-v1.5-7B.int8()                | 7518MiB |       | 35.2 | 61.87 | 68.22 | 350.71 | 1508.41 | 86.54 | 61.85 | 398ms ± 1.31ms         |
> | LLaVA-6B-layerwise+recovery         | 11604MiB| 15%   | 35.40| 61.17 | 68.07 | 328.57 | 1454.20 | 86.51 | 60.82 | 95ms ± 8.1ms           |
> | LLaVA-6B-layerwise+recovery.int8()  | 6473MiB | 15%   | 35.40| 61.17 | 68.07 | 328.57 | 1454.20 | 86.51 | 60.82 | 125ms ± 937μs          |
> | LLaVA-5B-widthwise+recovery         | 9548MiB | 30%   | 31.80| 60.71 | 60.54 | 252.50 | 1407.08 | 86.68 | 56.94 | 80.7ms ± 634μs         |
> | LLaVA-5B-widthwise+recovery.int8()  | 5389MiB | 30%   | 31.6 | 60.65 | 60.09 | 263.57 | 1410.28 | 86.78 | 57.10 | 141ms ± 2.4ms          |
>
> Unstructured pruning is not included in our evaluation as it provides limited memory saving, especially for edge devices, and requires specialized hardware and software for speed-up [1]. In contrast, structured pruning removes entire groups of parameters (e.g., neurons or layers), enabling significant memory reductions and making it better suited for resource-constrained environments like edge deployment.
>
> Overall, we want to highlight our study's aim to identify best practices for compressing MLLMs using structured pruning and efficient recovery methods. We believe that our findings can offer valuable guidelines for deploying MLLMs in resource-limited environments.

---

> ### Author Response · Authors · 2024-11-22
> **Author Response to Reviewer T1Q7**
>
> >While the framework demonstrates potential at lower compression ratios, its performance gains are limited at higher compression ratios. This limitation may reduce the practical appeal of the method for applications that demand more aggressive compression while still preserving task performance.
>
> We acknowledge that performance degradation at high compression ratios is a significant challenge. However, our study focuses on establishing best practices for compressing MLLMs through structured pruning, with a particular emphasis on the trade-off between model size reduction and performance retention. To this end, we systematically investigate a wide range of compression ratios, a scope often overlooked in prior work [2]. Our primary goal is to provide practitioners with a comprehensive framework to make informed decisions about compression strategies, rather than solely aiming to optimize performance at extreme compression levels.
>
> >Although the proposed combination of multiple pruning and recovery strategies is thorough, it increases implementation complexity.
> We acknowledge the reviewer’s concern about the implementation complexity. While our paper presents a broad comparison of pruning and recovery strategies, our aim is to provide an in-depth evaluation of different methods, highlighting their strengths and weaknesses so that practitioners can select the most suitable approach for their specific needs without conducting extensive experiments themselves.
>
> >The reliance on complex data structures and recovery steps may hinder deployment. An author should clarify this.
>
> Our analysis does not rely on complex data structures. To simplify deployment in data-constrained scenarios, we use only a small fraction of the original training dataset for recovery training (see Figure 4). Specifically, our study shows that practitioners can achieve 95% performance recovery using just 5% of the original dataset, significantly reducing the reliance on large datasets and making the process more practical for real-world applications.
>
> >What are the limitations of the chosen pruning strategies for different types of tasks within MLLMs?
>
> The weakness of both pruning strategies is the limited performance in the high compression ratio scenarios. As shown in Table 6, both pruning methods lead to significant performance degradation across all benchmarks at high pruning ratios. Notably, when only the multimodal projector is finetuned, the layerwise pruned model recovers substantial performance, suggesting that layerwise pruning primarily harms the alignment between visual and textual features.
>
> In practice, the implementation requirements for the two pruning methods differ. Widthwise pruning requires gradient information, leading to higher memory demands, while layerwise pruning has lower memory requirements as it relies only on activations.

---

> ### Author Response · Authors · 2024-11-22
> **Author Response to Reviewer T1Q7**
>
> >Can the proposed data-efficient recovery techniques be generalized to other model architectures or require specific adjustments?
>
> We appreciate the reviewer’s question regarding the generalizability of our best practices. The techniques, indeed, generalize to other architectures as shown by our new experiments with the Mini-InternVL-Chat-4B-V1-5 [3] model. We added it to the revision in Appendix I. The results confirm that our observations hold for this model, demonstrating the applicability of our best practices to other MLLMs. Notably, all pruned models are recovery trained on only 3% of the original dataset, which highlights the data efficiently.
>
> In the table below, we observe that widthwise pruning continues to offer better performance without recovery training. With InternVL, widthwise pruning retains 97.4% of the model’s original performance at a 15% compression ratio, compared to 96.7% for layerwise pruning, reinforcing its suitability as a default strategy in low-resource scenarios.
>
> | Model                               | Size | Ratio | MMMU  | GQA   | SQA   | MME-C | MME-P | POPE | AVG   | AVG-% |
> |-------------------------------------|------|-------|-------|-------|-------|-------|-------|------|-------|--------|
> | Mini-InternVL-Chat-4B-V1-5         | 4B   |       | 43.20 | 62.57 | 93.30 | 547.50 | 1,596.71 | 88.00 | 72.56 | 100%   |
> | Layerwise (Prune Only)              | 3.5B | 15%   | 42.70 | 54.43 | 92.96 | 527.86 | 1,534.37 | 88.09 | 70.15 | 96.68% |
> | Widthwise (Prune Only)              | 3.5B | 15%   | 43.60 | 56.35 | 93.12 | 510.10 | 1,588.30 | 87.96 | 70.70 | 97.44% |
>
> Additionally, we find that finetuning only the multimodal projector is sufficient at small compression ratios, where pruning minimally impacts the language model but disrupts multimodal alignment. With InternVL, finetuning only the projector recovers 96.9% of the performance at a 15% compression ratio, compared to 97.8% when both the projector and language model are finetuned. At a 30% compression ratio, projector-only finetuning recovers 75.1% while finetuning both components recovers 86.6%. These results reinforce our observations from earlier models and validate the transferability of our proposed practices.
> | Model                               | Size | Ratio | MMMU  | GQA   | SQA   | MME-C | MME-P | POPE | AVG   | AVG-% |
> |-------------------------------------|------|-------|-------|-------|-------|-------|-------|------|-------|--------|
> | Mini-InternVL-Chat-4B-V1-5         | 4B   |       | 43.20 | 62.57 | 93.30 | 547.50 | 1,596.71 | 88.00 | 72.56 | 100%   |
> | Layerwise Prune + Finetuning mm-projector | 3.5B | 15%   | 42.70 | 54.43 | 92.96 | 527.86 | 1,534.37 | 88.09 | 70.15 | 96.68% |
> | Layerwise Prune + Finetuning mm-projector | 3B   | 30%   | 33.30 | 27.39 | 62.82 | 197.14 | 845.60  | 73.20 | 43.94 | 60.56% |
>
>
> | Model    | Size | Ratio | MMMU  | GQA   | SQA   | MME-C | MME-P | POPE | AVG   | AVG-% |
> |-------------------------------------|------|-------|-------|-------|-------|-------|-------|------|-------|--------|
> | Mini-InternVL-Chat-4B-V1-5         | 4B   |       | 43.20 | 62.57 | 93.30 | 547.50 | 1,596.71 | 88.00 | 72.56 | 100%   |
> | Layerwise Prune + Finetuning mm & LLM  | 3.5B | 15%   | 43.10 | 56.34 | 93.36 | 524.64 | 1,585.83 | 88.10 | 70.96 | 97.80% |
> | Layerwise Prune + Finetuning mm & LLM | 3B   | 30%   | 34.40 | 53.46 | 76.80 | 432.27 | 1,438.28 | 86.57 | 62.86 | 86.64% |
>
> Moreover, our initial findings show that the combination of supervised finetuning with intermediate representation distillation consistently yields the highest performance across compression ratios. With InternVL, this combined approach achieves 98.2% recovery at a 15% compression ratio and 87.2% at a 30% compression ratio, confirming its effectiveness.
>
> | Model | Size | Ratio | MMMU | GQA | SQA | MME-C | MME-P | POPE | AVG | AVG-% |
> |-------------------------------------|------|-------|-------|-------|-------|-------|-------|------|-------|--------|
> | Mini-InternVL-Chat-4B-V1-5         | 4B   |       | 43.20 | 62.57 | 93.30 | 547.50 | 1,596.71 | 88.00 | 72.56 | 100%   |
> | Layerwise Prune + Finetuning + Distillation | 3.5B | 15%   | 43.30 | 56.17 | 93.41 | 539.64 | 1,582.58 | 87.89 | 71.23 | 98.16% |
> | Layerwise Prune + Finetuning + Distillation | 3B   | 30%   | 36.20 | 53.77 | 76.60 | 448.21 | 1,410.88 | 86.60 | 63.29 | 87.23% |
>
> Overall, the results from InternVL indicate that our methods generalize well to newer MLLM architectures. We have incorporated these findings in the updated version of our paper to strengthen our contributions.
>
> Reference
>
> [1] Isaac–Chassande et al. "Dedicated hardware accelerators for processing of sparse matrices and vectors: a survey." ACM 2024: 1-26.
>
> [2] Ma et al. "Llm-pruner: On the structural pruning of large language models." NeurIPS 2023.
>
> [3] Chen et al. "Internvl: Scaling up vision foundation models and aligning for generic visual-linguistic tasks." CVPR 2024.

---

> > ### Comment · Reviewer_T1Q7 · 2024-11-24
> >
> > Thanks for addressing all my concerns. I'll maintain my score.

---

> > > ### Author Response · Authors · 2024-11-26
> > > **Author Response to Reviewer T1Q7**
> > >
> > > Thank you for getting back to us. We're glad we could address all your concerns. Do you have any other open questions? We're happy to reply.

---

> > > ### Author Response · Authors · 2024-11-30
> > > **Author Response to Reviewer T1Q7**
> > >
> > > Dear Reviewer T1Q7,
> > >
> > > We sincerely appreciate your thoughtful and constructive feedback. In response, we conducted additional experiments to combine quantization with pruning, demonstrating the complementarity of these techniques as effective methods for model compression. Moreover, we extended our best practices to the InternVL model, showcasing the generalizability of our approach to newer MLLM architectures.
> > >
> > > Our paper aims to offer actionable insights and practical techniques for MLLM compression through pruning and knowledge distillation, helping practitioners save both time and computational resources. We are glad to have addressed all your concerns and provided further evidence to strengthen our contributions.
> > >
> > > In light of these enhancements, we kindly ask you to reconsider and potentially adjust your review score, taking into account the improvements and additional evidence provided during the rebuttal. Thank you once again for your valuable feedback and for considering our request.

---

### Author Response · Authors · 2024-11-22
**Global Author Response**

We thank the reviewers for their thoughtful feedback, suggesting several improvements to our work.
We are glad that the reviewers recognized our contributions listed below:
* This paper provides extensive experimentation across two MLLMs with varying compression ratios, recovery techniques, and data requirements (T1Q7 and J6JH).
* The experiments are well-designed and cover multiple dimensions: pruning methods, and data efficiency (J6JH). This paper provides a detailed view of how different configurations affect model performance (T1Q7) and concrete best practices for practitioners (zH8E).
* This paper emphasizes data-efficient model recovery, highlighting scenarios where only 5% of the original data suffices to restore a substantial portion of the model’s performance (T1Q7).

In response to the reviewer’s comments, we conducted new experiments and analyses to address the raised concerns and further strengthen our contributions.

* Combining quantization with pruning. To expand the scope, we included 8-bit quantization experiments for the LLaVA model and its pruned variants. These experiments demonstrate that quantization reduces the memory footprint by up to 44.5% with minimal performance loss (0.43 percentage points), while combining quantization with structured pruning achieves further reductions of 40–44% at compression ratios of 15% and 30%, with performance losses of only 0.4 and 1.3 percentage points(pp). These findings highlight the complementarity of quantization and pruning as effective compression techniques.

* New MLLM models. We extend our study to the Mini-InternVL-Chat-4B-V1-5 model, confirming the generalizability of our methods. Widthwise pruning consistently outperforms layerwise pruning without recovery training, retaining 97.4% of original performance at a 15% compression ratio. Like the previous model, InternVL benefits from finetuning the projector at small compression ratios, but finetuning both the projector and LLM remains necessary for larger ratios. Additionally, incorporating a distillation loss on intermediate features consistently enhances recovery results, mirroring earlier findings.

* Revised presentation. We also revised the related work section to include recent studies and clarified the distinct focus of our work on compressing existing MLLMs rather than building efficient models from scratch. Finally, we emphasize the practicality of our methods for deployment, showcasing the data efficiency of our recovery strategies, which use only a fraction of the original training data.

In summary, our study systematically evaluates structured pruning and recovery methods for MLLMs, addressing unique challenges like modality misalignment and providing best practices for resource-constrained deployments. We believe these additions further strengthen the contributions of our work.

We would be happy to address any additional comments from the reviewers in the rest of the rebuttal period.

---

### Comment · Area_Chair_88XX · 2024-11-24

Dear Reviewers,

This is a friendly reminder that the discussion period will end on Nov 26th (Anywhere on Earth). If you have not already, please take a careful look at the other reviews and author responses, and comment on whether your original rating stands. Thank you.

Best, AC

---

### Comment · Area_Chair_88XX · 2024-11-28

Dear reviewers,

This is a friendly reminder that the discussion period has been extended until December 2nd. If you haven’t yet, we kindly encourage you to review the authors' rebuttal and messages at your earliest convenience and confirm whether your comments have been adequately addressed.

We greatly appreciate your service to this process.

Best, AC

---

### Meta-Review · Area_Chair_88XX · 2024-12-19

**Metareview:**

This paper investigates efficient compression techniques for MLLMs, focusing on two key pruning strategies, width and layerwise.  The paper received scores of 6,3,5.  Mentioned strengths include extensive and well-designed experiments, and the emphasis on data-efficient model recovery.  Mentioned weaknesses include limited novelty and technical contribution, some overclaimed statements, and missing important experimental comparisons.  The rebuttal and discussion by the authors included new experiments (including combining quantization with pruning, and more MLLM models) and revised presentation.  While the rebuttal and discussion addressed some concerns, other including limited novelty and technical contribution remained.  After carefully considering the paper, rebuttal, and discussion, the AC does not feel that the paper is ready for acceptance to ICLR.

**Additional Comments On Reviewer Discussion:**

Mentioned strengths include extensive and well-designed experiments, and the emphasis on data-efficient model recovery.  Mentioned weaknesses include limited novelty and technical contribution, some overclaimed statements, and missing important experimental comparisons.  The rebuttal and discussion by the authors included new experiments (including combining quantization with pruning, and more MLLM models) and revised presentation.  While the rebuttal and discussion addressed some concerns, other including limited novelty and technical contribution remained.  After carefully considering the paper, rebuttal, and discussion, the AC does not feel that the paper is ready for acceptance to ICLR.

---

### Decision · Program_Chairs · 2025-01-22

Reject